# LocCa: Visual Pretraining with Location-aware Captioners

**Bo Wan**$_{1,3}$ *    **Michael Tschannen**$_1$    **Yongqin Xian**$_2$    **Filip Pavetic**$_1$
**Ibrahim Alabdulmohsin**$_1$    **Xiao Wang**$_1$    **André Susano Pinto**$_1$
**Andreas Steiner**$_1$    **Lucas Beyer**$_1$    **Xiaohua Zhai**$_1^{\dagger}$

$^1$Google DeepMind, Zürich    $^2$Google, Zürich    $^3$KU Leuven

## Abstract

Image captioning was recently found to be an effective pretraining method similar to contrastive pretraining. This opens up the largely-unexplored potential of using natural language as a flexible and powerful interface for handling diverse pretraining tasks. In this paper, we demonstrate this with a novel visual pretraining paradigm, LocCa, that incorporates location-aware tasks into captioners to teach models to extract rich information from images. Specifically, LocCa employs two tasks, bounding box prediction and location-dependent captioning, conditioned on the image pixel input. Thanks to the multitask capabilities of an encoder-decoder architecture, we show that an image captioner can effortlessly handle multiple tasks during pretraining. LocCa significantly outperforms standard captioners on downstream localization tasks, achieving state-of-the-art results on RefCOCO/+/g, while maintaining comparable performance on holistic tasks. Our work paves the way for further exploration of natural language interfaces in visual pretraining.

## 1 Introduction

Remarkable progress has been made in large-scale visual pretraining [1, 2, 3, 4], where vision models are pretrained on large-scale annotated datasets [5, 6, 4] with a supervised classification loss. Yet, the manual annotation required for such datasets is time-consuming and costly, posing a challenge to scalability.

In light of this, the modern contrastive pretraining methods [7, 8] extract learning signals from web-crawled image-text pairwise datasets [9, 10], circumventing the need for extensive manual annotations. The contrastively pretrained models demonstrate remarkable capability on zero-shot transfer tasks, especially on downstream applications that require fine-grained visual [11, 12] or textual understanding [13]. More recently, image captioning has been shown as an alternative visual pretraining task to learn capable vision encoders [14], where an encoder-decoder architecture is pretrained to generate text captions from the image input. Some studies, such as [15, 16], pioneered the joint pretraining of contrastive and generative methods. Typically, encoded image features are fed into two parallel branches: one employs a text encoder to produce sentence embeddings for contrastive learning, while the other utilizes a text decoder to generate image captions. Despite the effectiveness of these works, they typically focus on a holistic understanding of images, often overlooking the region-specific details of the visual content.

The recent success of image captioning [14] and the advancements in multitasking learning of decoders [17, 18, 19] opens up the largely-unexplored potential of using natural language as a

---

*Work done during Google DeepMind internship. † Project lead. All authors made significant technical contributions.

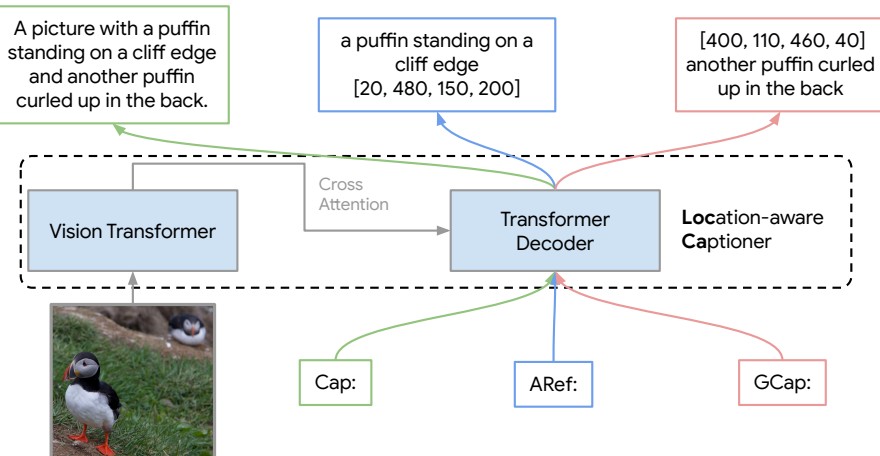

Figure 1: **Overview of LocCa**. LocCa consists of a standard vision transformer and a transformer decoder. The vision transformer takes image pixel as input, produces visual tokens as cross attention input to the transformer decoder. The transformer decoder is trained to read out rich information from the visual tokens. We adopt the following three task for pretraining: Cap, AREF and GCAP.

flexible and powerful interface for handling diverse tasks. We demonstrate this with a novel visual pretraining paradigm, LocCa, that enhances the visual representation with location-aware context. Works including [20, 21, 22] investigate the matching of image regions with corresponding text during pretraining. The central concept involves extracting Region of Interest (RoI) features from image embedding to facilitate contrastive learning with corresponding textual features. These approaches yield encouraging outcomes in location-sensitive tasks, such as object detection [23, 24, 25, 26] and referring expression [27, 28, 29, 30]. However, they require complex model architectures (e.g. RPN [24] and FPN [25]) for RoI generation. Also, given the presence of multiple object candidates within an image, region-wise matching becomes computationally demanding.

By contrast, LocCa is a simple yet effective location-aware captioning pretraining method as shown in Figure 1, which uses an autoregressive decoder as an interface to handle additional location-aware pretraining tasks. Concretely, other than the image-to-text captioning task, LocCa also pretrains the model with two location-aware tasks: (i) *automatic referring expressions*, which amounts to predict bounding box coordinates from automatically generated captions for specific image regions, and (ii) *grounded captioning* to jointly predict box coordinates and captions from the image. Specifically, LocCa leverages a multi-task decoder [17] for pretraining, where the model outputs are conditioned on the task prefixes for each task. Thanks to the shared vision transformer for multiple tasks, the additional localization losses are relatively cheap to compute, while the model inference speed is identical to the standard image caption pretrained models.

Our experimental results show that the LocCa performs significantly better on downstream tasks that require localization capabilities, while maintaining the same level of capabilities on holistic tasks. We summarize our contributions as follows: (i) For the first time we explore location-aware tasks as proxies for generative visual pretraining (as opposed to transfer/instruction tuning in prior works), enabling flexibly customized inference (detailed in Sec.3.3); (ii) Without bells and whistles, LocCa achieves state-of-the-art results on localization tasks, while preserving the competitive performance on holistic tasks; and (iii) When integrated in vision-language models, i.e. PaLI-3 [31], the vision encoder outperforms strong SigLIP baselines [13].

## 2   Related Works

Contrastive visual pretraining is a prominent direction in training vision and vision-language foundation models. Early works [32, 33, 34, 35] explore image-only contrastive loss by matching different views of the same image in the self-supervised learning setting. In vision-language model pretraining, CLIP [7] and ALIGN [8] show that a two-tower architecture trained with the contrastive objective on noisy image-text pairs can learn highly transferable image and text representations for various

downstream tasks. There have been many follow-up works [10, 36, 13, 37] that further improve the zero-shot image classification and retrieval performance. Notably, [20, 21] propose to incorporate location cues by contrastively aligning image regions and text phrases. In contrast, our work focuses on learning a location-aware vision-language model with a generative loss.

A natural alternative to contrastive pretraining is image captioning: Rather than matching image and text embeddings, one tries to predict captions from an image embedding. Early works investigate this approach at small scale [38, 39, 40, 41]. Later works augment large-scale contrastive pretraining with a captioning loss [15, 16, 42], or scale-up captioning as a stand-alone task without investigating transfer to a broad set of vision tasks [43, 44]. [14] recently showed that image captioning alone leads to visual representations competitive with contrastive pretraining.

Many recent large multimodal models are trained with a mixture of tasks [45, 18, 19, 46, 10, 47, 31, 48, 49]. Among the most popular types of tasks are those which can be formulated by mapping an image to a text string, including captioning, detection, and VQA [50, 45, 18, 19, 10, 47, 31, 48]. Another popular group of tasks are dense prediction tasks such as segmentation and depth prediction [19, 46]. While several studies have enhanced model pretraining by incorporating location information, their methodologies primarily leverage either pretrained language decoders [18, 51, 52] or pretrained cross-modal encoder-decoder [51] to integrate vision and language features for multitasking purposes, often neglecting the independent significance of visual pretraining from scratch. Furthermore, there is a trend of towards co-training on images, video, and audio [53, 54, 55], highlighting the multifaceted nature of current multi-modal research. Crucially, essentially all of these works rely on pretrained vision and language backbones, and merely fine-tune those together on the described tasks. Here, by contrast, we use multi-task pretraining to train visual encoders from scratch.

## 3 Location-aware Captioner

In this section, we introduce the location-aware image captioner LocCa for multitask pretraining. LocCa builds on an image captioner but provides a recipe for integrating location-aware information during model pretraining.

### 3.1 Pretraining tasks

The pretraining phase of LocCa draws inspiration from pioneering works that have successfully integrated a unified decoder for multitasking based on pretrained models [18, 10, 19, 46, 52], utilizing a task-specific prefix for each distinct task. This enhances the model's ability to handle multiple tasks concurrently.

For conventional image captioning, the process involves taking an image $x$ as input and generating a sequence of text tokens $y = [y_1, y_2, \ldots, y_n]$. In the LocCa framework, a task-specific prefix, labeled as "*Cap*:", is added to the beginning of the caption sequence to designate the task at hand. Moreover, LocCa integrates two additional location-aware tasks during its pretraining phase: *automatic referring expression* (AREF) and *grounded captioning* (GCAP). These tasks are inspired by referring expression comprehension [27, 28, 29, 30] and dense captioning [56, 57, 58, 59, 60] respectively. The key difference is that LocCa predicts both regional captions and box coordinates sequentially with task prefixes, instead of relying on either caption or box conditional inputs (see Fig. 1).

The foundation of LocCa's pretraining is built upon dense, automatically generated region annotations. Each image $x$ is associated with a comprehensive set of annotations $\{(b, c)\}$, where $b \in \mathcal{N}^4$ denotes the bounding box coordinates, and $c$ represents the corresponding textual descriptions or labels. For every bounding box, two distinct prompts are generated to cater to the aforementioned location-aware tasks: "*ARef*: $\{c\}$ : $\{b\}$" for automatic referring expression and "*GCap*: $\{b\}$ : $\{c\}$" for grounded captioning, each prefixed with "*ARef*:" and "*GCap*:" respectively. These prompts are then tokenized to produce the sequence $y$ for each task, facilitating pretraining with a text interface.

For each image, LocCa utilizes the same visual features extracted by the image encoder and performs three tasks using the same decoder in parallel. This pretraining scheme aims to make LocCa adept at linking fine-grained regional visual elements with appropriate textual descriptions.

## 3.2 Model details

**Architecture**   LocCa utilizes a conventional encoder-decoder framework, where the encoder comprises a Vision Transformer [2] to transform the input image $x$ into a sequence of feature embeddings. The decoder, built on a Transformer architecture [61], processes these image features, employing cross-attention across each layer to integrate visual and textual information effectively.

**Autogressive decoding**   In the decoding stage, LocCa uses causal attention masks [61] to guide the prediction of each token in the sequence, ensuring that each token is generated based on the ones before it, in a step-by-step manner. This setup helps in creating coherent and context-aware captions, drawing from the visual cues encoded earlier and maintaining a logical flow in the generated text.

**Parallel prediction**   Inspired by [14], LocCa also adopts parallel prediction for a fraction of the training examples (concretely 50%) in the standard image captioning task. This technique requires the model to predict the caption tokens independently in parallel, focusing exclusively on visual information to prevent the model from relying on preceding text tokens to predict the following ones. This strategy was shown to improve the learned representation on a range of tasks [14].

**Objective**   The optimization of LocCa's parameters $\theta$ is achieved through the maximization of the log-likelihood: $\sum_{i=1}^{|y|} \log P_\theta(y_i|y_{<i}, x)$. It is important to emphasize that, in the learning process for the location-aware tasks, LocCa is structured to predict captions and bounding boxes sequentially, contrasting with traditional approaches that might predict a caption based on a given bounding box or vice versa [18, 51]. This is achieved by applying losses to the entire prompt excluding the task prefix. Taking the AREF task as an example, the overall loss is computed for both textual predictions $c$ and box coordinates $b$. The loss on $c$ guides the model to identify a foreground region and generate its caption, while the loss on $b$ aims at refining the model's ability to accurately regress the box location relative to the caption.

## 3.3 Discussion

To the best of our knowledge, LocCa is the first end-to-end method that incorporates the location-aware tasks (i.e., AREF and GCAP) into generative VLM *pretraining*. Our key novelty lies in the formulation of the location-aware proxy tasks which allows for scalable pretraining and enhances the visual localization capabilities. Compared to [22, 18, 51, 52] that require either a pretrained visual encoder or language decoder, LocCa does not need any pretrained model for initialization. Compared to [22, 62] that employ complex architectures with multiple losses, LocCa adopts a simple encoder-decoder architecture with a single generative loss. Compared to [18, 19] that are pretrained on a large number of tasks, LocCa introduces the novel use of simple AREF and GCAP proxy tasks for visual pretraining.

The dual-faceted loss structure of AREF and GCAP achieves comparable results to directly adopting the traditional referring expression and dense captioning tasks. However, it enhances inference flexibility of LocCa, allowing for varied input configurations. For instance, users can input a single task prefix (e.g., "*ARef*:") to prompt the model to identify and describe an area of interest along with its location. Alternatively, by inputting both the task and a conditional input (e.g., "*ARef*: a black and white cat :"), LocCa can be directed to focus solely on predicting the location. This flexibility allows for customized responses to various inquiries, highlighting the model's adaptability to meet specific user needs.

## 4 Experiments

### 4.1 Experimental setup

**Pretraining dataset**   We use a subset of the WebLI dataset [10] corresponding to English websites and apply text-based filtering [8] to obtain 1B image/alt-text pairs. The WebLI data was de-duplicated w.r.t. all the images in the evaluation data sets used in this paper. To obtain fine-grained object locations, a publicly available OWL-ViT CLIP L/14 model [63] is applied to generate detection pseudo annotations. Specifically, two groups of box categories are generated: the n-gram texts from alt-text and the object categories as used by PaLI [10], more details can be found in [64]. In LocCa pretraining, we first filter the candidate bounding boxes according to their OWL-ViT confidence score

with a threshold of 0.3, and then randomly sample one box-caption or box-category pair for referring expression and grounded captioning separately.

**Baselines**     Our main baselines are CLIP-style contrastively pretrained dual-encoder models [7, 8] on our dataset (referred to as CLIP* to differentiate from the model released by [7]), as well as the captioning-pretrained encoder-decoder models from [14]. For both types of models we follow the exact training recipe from [14]. For the captioning-based pretraining we consider standard autoregressive captioning (Cap) as well as the variant with parallel prediction (CapPa). This variant removes the decoder attention mask for a fraction of the training examples and replaces the decoder input, which corresponds to the right-shifted output sequence during autoregressive training, with a sequence of all mask tokens. Parallel prediction [14] improves the representation learning capabilities captioning-based pretraining at no extra computation cost.

**Implementation details**     LocCa adopts an encoder-decoder structure where, unless otherwise noted, the encoder defaults to a standard ViT-L/14 design with 24 transformer blocks handling input image patches of size 14. The decoder, following [14], is a Transformer-L model consisting of 12 transformer decoder blocks. In total, the model comprises approximately 600M parameters. Two more LocCa setups with ViT-B/16 and ViT-G/14 (see [4] for model specifications) are adopted for ablation tests and scaling experiments respectively.

**Pretraining details**     LocCa is pretrained for about 9 billion image/alt-text seen examples, which corresponds to about 9 epochs on our tailored subset of WebLI. For the optimizer, we employ the Scaling-ViT AdaFactor variant [4], combined with a cosine schedule that includes 10,000 warmup steps. The batch size is set at 8,192, while the learning rate and decay factor are adjusted to $10^{-3}$ and $10^{-4}$, respectively. During this process, images are uniformly resized to a resolution of 224 x 224 pixels. Alt-texts are tokenized into a vocabulary consisting of 32,000 tokens using a sentence piece model trained on the English segment of C4 [65], with a cap on the sequence length at 64 tokens. We represent bounding box coordinates using up to 500 integral numbers, which are then directly converted into strings for straightforward representation of coordinate tokens. For parallel prediction in the vanilla image captioning task, the fraction of examples is set to 50% by default. The pretraining of $\text{LocCa}_L$ takes 153 hours using 256 TPUv3 chips.

**Downstream tasks**     Our goal with LocCa is to preserve or even improve the capability on various image-level understanding tasks, and get a higher performance on fine-grained location-aware tasks. To this end, we assess the capabilities of LocCa in holistic and location-aware image understanding tasks across a range of downstream tasks. In the realm of holistic image understanding, we focus on the same LiT-Decoder tasks [17] in CapPa [14] including image classification (CLS)[5, 66, 67, 68, 69], image captioning (CAP) [70, 71], optical character recognition (OCR-VQA) [72], visual question answering (VQA) [73], and graph question answering (GQA) [74]. For evaluating location-aware image understanding, we choose two widely recognized tasks like referring expression comprehension (REC) [27], referring expression segmentation (RES) [75] and object detection (OD) [24]. Additionally, paralleling the approaches of PaLI [10, 31], we integrate LocCa to a pretrained large language model to assess performance on a variety of vision-language tasks, including image captioning and visual question answering. Notably, despite LocCa not being exposed to any video content during pretraining, we adapt it to various video-related tasks, such as video captioning, QA, and classification. This adaptation aims to assess its generalization capabilities to new modalities, demonstrating its versatility compared to other image-text pretraining approaches. We adopt different strategies for transferring LocCa to downstream tasks, as summarized in Appendix A.1.

## 4.2   Quantitative results

We conduct extensive experiments to evaluate LocCa. The integration of location-aware cues enables LocCa to maintain its performance on holistic image understanding tasks while achieving substantially improved outcomes on location-aware tasks. Further enhanced by an advanced pretrained large language model, LocCa exhibits exceptional performance across a range of vision-language tasks, substantially surpassing baseline models.

**Referring Expression Comprehension**     In this section, we present results on the referring expression comprehension benchmarks, including RefCOCO, RefCOCO+ and RefCOCOg [28]. As shown in Table 1, LocCa establishes a new state-of-the-art across these benchmarks. This advancement is particularly noteworthy considering the inherent limitations of previous methods. For example,

Table 1: Result comparison with previous SOTA methods on RefCOCO benchmarks.

| MODEL | RefCOCO | | | RefCOCO+ | | | RefCOCOg | |
|---|---|---|---|---|---|---|---|---|
| | val | testA | testB | val | testA | testB | val-u | test-u |
| *Models that trained on val/test images, see text for more.* | | | | | | | | |
| PixelLLM[52] | 89.8 | 92.2 | 86.4 | 83.2 | 87.0 | 78.9 | 84.6 | 86.0 |
| UniTAB[76] | 88.59 | 91.06 | 83.75 | 80.97 | 85.36 | 71.55 | 84.58 | 84.70 |
| $OFA_L$[18] | 90.05 | 92.93 | 85.26 | 85.80 | 89.87 | 79.22 | 85.89 | 86.55 |
| $UNINEXT_L$[77] | 91.43 | 93.73 | 88.93 | 83.09 | 87.90 | 76.15 | 86.91 | 87.48 |
| $LocCa_L$ | **91.94** | **94.56** | **89.13** | **86.47** | **91.67** | **80.43** | **87.46** | **87.95** |
| $OFA_H$[18] | 92.04 | 94.03 | 88.44 | 87.86 | 91.70 | 80.71 | 88.07 | 88.78 |
| $UNINEXT_H$[77] | 92.64 | 94.33 | **91.46** | 85.24 | 89.63 | 79.79 | 88.73 | 89.37 |
| ONE-PEACE$_{1.5B}$[62] | 92.58 | 94.18 | 89.26 | 88.77 | 92.21 | 83.23 | 89.22 | 89.27 |
| Shikra$_{13B}$[78] | 87.83 | 91.11 | 81.81 | 82.89 | 87.79 | 74.41 | 82.64 | 83.16 |
| Ferret$_{13B}$[79] | 89.48 | 92.41 | 84.36 | 82.81 | 88.14 | 75.17 | 85.83 | 86.34 |
| $LocCa_G$ | **92.99** | **95.02** | 90.48 | **89.12** | **92.87** | **83.55** | **89.24** | **89.90** |
| *Models that have seen val/test images during pre-training, see text for more.* | | | | | | | | |
| $UNITER_L$[80] | 81.41 | 87.04 | 74.17 | 75.90 | 81.45 | 66.70 | 74.86 | 75.77 |
| $VILLA_L$[81] | 82.39 | 87.48 | 74.84 | 76.17 | 81.54 | 66.84 | 76.18 | 76.71 |
| MDETR[82] | 86.75 | 89.58 | 81.41 | 79.52 | 84.09 | 70.62 | 81.64 | 80.89 |
| *Models that have never seen val/test images, see text for more.* | | | | | | | | |
| RefTR[83] | 85.65 | 88.73 | 81.16 | 77.55 | 82.26 | 68.99 | 79.25 | 80.01 |
| $LocCa_L$ | 89.70 | 92.75 | 85.30 | 83.85 | 89.40 | 76.76 | 84.62 | 85.86 |
| $LocCa_G$ | **91.20** | **93.34** | **87.56** | **86.89** | **90.71** | **80.73** | **87.34** | **87.90** |

UNINEXT [77], which adopts a Deformable DETR architecture [84] tailored for detection-based tasks, and OFA, which requires a pretrained BART [85] for initialization, both achieve good results but are constrained by their specialized setups. Besides, some location-aware VLMs, such as Shikra [78] and Ferret [79], require an LLM for knowledge-based reasoning, which increases inference costs. In contrast, LocCa employs a standard encoder-decoder architecture with auto-regressive pretraining from scratch, significantly outperforming these methods across all benchmarks without the need for complex task-specific adaptations.

Achieving good performance on RefCOCO usually requires pretraining at high image resolutions. For instance, $OFA_L$ is pretrained at $480^2$px, while UNI-NEXT undergoes multi-scale pre-training. We opt for a standard $224^2$px pretrain resolution for simplicity. We transfer the $224^2$px pretrained model to RefCOCO by fine-tuning with $640^2$px resolution, using learning rate $10^{-4}$ and no weight decay. We report the standard metric Acc@0.5 on the validation and test sets.

Notably, we train on the combined training sets of RefCOCO, RefCOCO+ and RefCOCOg but *removing all validation and test images* from this combination. The splits of these three datasets are largely overlapping, meaning that methods trained on the combined training sets without de-duplication, a recently common phenomenon, have trained on about half of the test images, see Appendix A.3 for details. We provide the list of removed image IDs in the Appendix A.3. Furthermore, some models such as UNITER [80], VILLA [81], and MDETR [82] use COCO pre-trained detection components which have seen test images from the three RefCOCO versions. We have removed all training, validation, and test images from the COCO dataset from our pre-training data, as well as near-duplicates thereof. Hence, we group models reported in Table 1 into three distinct categories. We transfer LocCa on both the full and the "clean" combined training set and provide both results. We hope that, going forward, the community evaluates models on RefCOCO in this clean setup.

Moreover, as shown in Table 2, LocCa improved significantly over all the baselines of image-text pretrained models. To ensure a fair comparison with contrastive baselines (i.e. CLIP) which lack a text decoder for box prediction, we employ the LiT-Decoder [17] setup for comparison on RefCOCO benchmarks. It involves freezing the pretrained image encoder while training a text decoder from scratch with a small resolution of $224^2$px. This setup is necessary to properly compare with CLIP-style models without a decoder. The performance improvements attributable to location-aware pretraining are evident, demonstrating the enhanced sensitivity of visual encoder to object regions, which is crucial for excelling in referring expression tasks.

Table 2: Comparison with baselines on RefCOCOs. A randomly initialized decoder is trained for REC and RES, with frozen image encoders. We report Acc@0.5 for REC and mIoU for RES. Here CLIP uses model checkpoints released by [7]; all other baselines use the same data as LocCa.

| | MODEL | RefCOCO | | | RefCOCO+ | | | RefCOCOg | |
|---|---|---|---|---|---|---|---|---|---|
| | | val | testA | testB | val | testA | testB | val-u | test-u |
| REC | CLIP [7] | 65.21 | 71.28 | 58.17 | 57.53 | 66.44 | 47.77 | 59.32 | 60.24 |
| | CLIP* | 58.28 | 63.59 | 53.73 | 49.01 | 55.94 | 41.96 | 55.70 | 55.88 |
| | Cap [14] | 60.64 | 65.47 | 56.17 | 52.56 | 58.32 | 45.99 | 56.75 | 57.99 |
| | CapPa [14] | 64.17 | 69.90 | 58.25 | 56.14 | 63.68 | 48.18 | 58.90 | 59.91 |
| | LocCa | **88.34** | **91.20** | **85.10** | **79.39** | **85.13** | **72.61** | **81.69** | **82.64** |
| RES | CLIP [7] | 36.15 | 38.25 | 35.82 | 31.40 | 36.00 | 28.69 | 29.21 | 29.44 |
| | CLIP* | 32.78 | 34.89 | 33.03 | 27.49 | 30.14 | 25.07 | 26.99 | 26.83 |
| | Cap [14] | 34.84 | 37.68 | 35.39 | 31.93 | 35.24 | 28.79 | 28.84 | 29.11 |
| | CapPa [14] | 36.62 | 39.23 | 36.67 | 32.54 | 37.49 | 29.59 | 30.40 | 30.43 |
| | LocCa | **64.98** | **65.39** | **64.09** | **57.85** | **60.92** | **52.72** | **55.84** | **56.95** |

Table 3: Results on holistic image understanding tasks. Here CLIP uses model checkpoints released by [7]; all other baselines are trained on the same data as LocCa.

| MODEL | Classification | | | | | Captioning | | OCR | VQA | |
|---|---|---|---|---|---|---|---|---|---|---|
| | i1k | sun | food | res | pet | COCO | Flickr | VQA | VQAv2 | GQA |
| CLIP [7] | **84.8** | 84.8 | **95.2** | 96.3 | 95.4 | 124.4 | 87.1 | 64.1 | 70.4 | 58.7 |
| CLIP* | 84.7 | **85.7** | 94.6 | **96.4** | 95.2 | 123.2 | 85.5 | 61.3 | 68.5 | 55.3 |
| Cap [14] | 83.8 | 84.7 | 93.4 | 95.1 | 95.0 | 125.9 | 88.3 | 64.2 | 70.9 | 58.5 |
| CapPa [14] | 84.4 | 84.9 | 93.8 | 96.0 | **95.6** | 125.8 | 89.3 | **65.6** | 70.9 | 58.3 |
| LocCa | 84.5 | 84.9 | 93.9 | 96.0 | 95.0 | **127.1** | **90.7** | 64.5 | **72.8** | **61.8** |
| LocCa$_G$ | 85.8 | 85.1 | 95.4 | 96.1 | 95.8 | 130.9 | 92.6 | 67.6 | 73.9 | 61.5 |

**Referring Expression Segmentation**    For referring expression segmentation, we extend the REC task by adding a suffix "*Mask*:" to the text, followed by the indexes of segmentation tokens. The segmentation tokens specify the precise shape of the segmentation mask within the bounding box that is identified during the REC task. This process leverages a pretrained VQ-VAE [86] to convert the semantic masks to tokens, please refer to Appendix A.3 for more details.

LocCa is adapted to RES under the same settings as in REC, using the "clean" combined training sets of RefCOCO, RefCOCO+, and RefCOCOg (c.f. Appendix C). Specifically, we compare different frozen encoders and train a decoder from scratch. As shown in Table 2, LocCa's vision encoder outperforms other encoders substantially, thereby providing further validation of its location-wise sensitivity. Moreover, we employ the full encoder-decoder LocCa model and fine-tune it for RES. Notably, LocCa achieves competitive results even compared to the state-of-the-art PaLI-3 model, albeit with considerably fewer parameters (0.6B vs. 5B). See Appendix A.3 for details.

**Holistic Image Understanding**    While being great on the referring expression comprehension tasks, we also verified that LocCa performs equally well on the holistic image understanding tasks. Interestingly, we found that LocCa outperforms the image-text pretrained baselines on the object-centric tasks like VQAv2 and GQA.

Following the similar evaluation protocol in [14], we evaluate the capability of LocCa with the "LiT decoder" [17] setup, to investigate the adaptation capability of the learned representations to interface with a text decoder. Here we report the classification accuracy on 5 classification (CLS) datasets (ImageNet-1k [5], Sun-397 [67], Food-101 [66], Resisc-45 [68], Oxford-Pet [69]), and also answer accuracy on VQAv2 and GQA and OCR-VQA datasets. Besides, we report the CIDEr score on 2 captioning (CAP) datasets (COCO [70] and Flickr [71]).

As shown in Table 3, the performance of LocCa is better than image-text pretrained baselines on image captioning, VQA and GQA, comparable on image classification, and slightly lags on OCR-

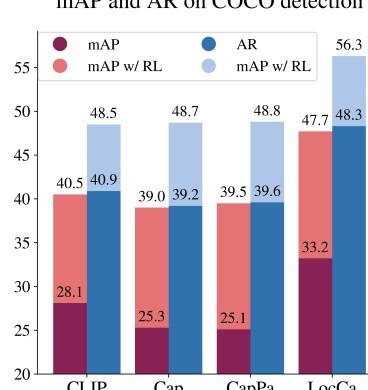

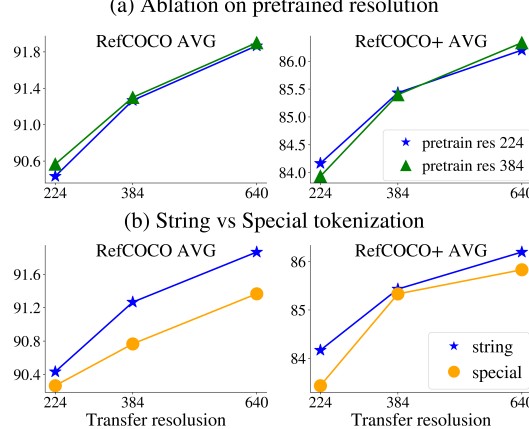

Figure 2: Result on COCO detection with a limit of 25 output boxes. For reward tuned models we show both the results before (dark blue and orange) and after (light blue and orange) reinforce tuning[88].

Figure 3: Ablation studies on (a) impact of different pretrained image resolutions on string token; and (b) string vs special token of box coordinates with pretrained res 224. The results are the average Acc@0.5 of the val&test splits on RefCOCO/+.

Table 4: Results on PaLI-3 [31] transfers to diverse captioning and question answering tasks. LocCa consistently outperforms other image encoders, especially on tasks requiring understanding of objects, including both natural and text (OCR) objects.

| **MODEL** | COCO | VQAv2 | OKVQA | TextVQA | ST-VQA | TallyQA |
|---|---|---|---|---|---|---|
| SigLIP$_L$ [13] | 135.8 | 75.6 | 57.5 | 41.1 | 46.2 | 74.9/61.4 |
| Cap$_L$ [14] | 135.0 | 75.3 | 57.5 | 44.6 | 44.5 | 73.2/62.0 |
| CapPa$_L$ [14] | 135.3 | 75.5 | 57.7 | 44.0 | 45.2 | 73.4/60.9 |
| LocCa$_L$ | **138.9** | **77.6** | **58.4** | **49.2** | **50.9** | **79.3/64.1** |
| SigLIP$_G$ | 140.3 | 77.5 | 58.6 | 50.6 | 50.5 | 76.3/61.8 |
| LocCa$_G$ | **140.9** | **78.1** | **59.8** | **52.1** | **52.3** | **79.0/64.2** |

VQA. Notably, both VQA and GQA necessitate fine-grained instance-level comprehension, focusing on the spatial and semantic relationships between objects to accurately provide the correct answer. For example, GQA involves complicated information about objects, attributes and relations provided by scene graphs [57]. Such an observation further reveals the advantage of LocCa on fine-grained object-level sensitivity.

It is also interesting to investigate the full potential of the LocCa model by fine-tuning it on holistic tasks with a small $3 \times 10^{-6}$ learning rate without weight decay. To this end, we choose the widely used COCO image captioning task as an example. LocCa achieves competitive results of 138.0 CIDEr score with a standard $224^2$px. When increasing the transfer resolution to $640^2$px, the LocCa$_L$ model achieves **140.3** CIDEr score without CIDEr metric optimization [87].

**Vision-Language Models with LocCa**   In this section, we present results on vision-language tasks with LocCa plugged into a pretrained large language model. More specifically, we use PaLI-3 [31] here to test the vision encoder quality. Importantly, we discovered that the generative LocCa vision backbone outperforms the SigLIP backbone, which is the default setup used in PaLI-3.

We select a wide range of tasks to assess the models' proficiency in understanding various visual concepts, including: image scene (COCO Caption), objects (VQAv2 and TallyQA [89]), visually-situated text (TextVQA [90], ST-VQA [91]) and general knowledge (OKVQA [92]). Remarkably, as shown in Table 4, LocCa consistently surpasses the Cap and CapPa vision encoders by a significant margin across all these tasks. Note that SigLIP$_L$ in Table 4 uses image patch size 16 while all the other models use patch size 14. SigLIP$_L$ is marked grey because it's not directly comparable. The LocCa$_G$ model consistently outperforms SigLIP$_G$ across all the tasks. With the knowledge of object and textual regions, LocCa is better positioned to understand more challenging and subtle

concepts like counting, visual relations, and word characters. Further fine-tuning details and the hyperparameters we used are provided in Appendix A.4.

**Object Detection**    To evaluate for object detection, we use COCO detection dataset and follow [88] which models the task with an encoder-decoder model that outputs sequences of bounding boxes similar to pix2seq[50]. The model is trained in two phases, first it maximizes the log-likelihood of generating the ground truth sequences of boxes, secondly it uses reinforce to tune the model for a reward related to the mAP metric.

Both [88, 50] pretrain their encoder-decoder model in Objects365 [93] and we found this to be critical. In particular the task output format is different from the ones LocCa used during pretraining and the size of COCO is too small to learn it without overfitting. To measure the performance of the encoders, we opt to initialize the decoder from an Objects365 pretrained model. More details of adapting LocCa for object detection please refer to Appendix A.5.

We present the comparative results in Figure 2, where LocCa significantly outperforms all image-text pretrained baselines in both mAP and AR, at both stages — before and after the application of reinforcement tuning. This performance aligns with our expectations regarding LocCa's object-centric comprehension, attributed to the integration of additional location-aware pretraining.

**Zero-shot Transfer**    Following locked-image tuning [36], we freeze the pre-trained vision encoder and train a text encoder contrastively to perform zero-shot classification and zero-shot retrieval on downstream tasks. With an L/14 architecture and 3B examples LiT-tune duration, $LocCa_L$ achieves 77.1% 0-shot accuracy on ImageNet, which is competitive to the same size $CapPa_L$ [14] model's 76.4% accuracy. On COCO retrieval tasks, $LocCa_L$ achieves 46.6% and 64.6% on text-to-image and image-to-text retrieval tasks respectively. It outperforms $CapPa_L$'s 43.9% and 60.3% COCO retrieval results by a large margin. Please refer to Appendix A.6 for more details.

**Semantic Segmentation**    We investigate the dense feature learning capabilities of LocCa by evaluating it for semantic segmentation on ADE20k [95] along with Cap and CapPa. To this end, we use the Segmenter framework [94] which attaches a linear layer to every patch embedding to predict the semantic label of that patch, and obtains the high-resolution semantic map by upsampling the low-resolution map. The results in Table 5 show that LocCa outperforms Cap and CapPa by about 2 mIoU points, and the Seg ViT-L baseline which is based on an ImageNet-21k-pretrained ViT-L by about 1 point. The transfer details are provided in Appendix A.7.

Table 5: Fine-tuning vision backbones with a linear head [94] for semantic seg. on ADE20k.

| MODEL | mIoU |
|---|---|
| Seg ViT-L [94] | 50.71 |
| Cap | $49.82^{\pm 0.6}$ |
| CapPa | $49.92^{\pm 0.1}$ |
| LocCa | $51.81^{\pm 0.3}$ |

**Video Understanding**    We follow PaLI-3 [31] to evaluate LocCa and CapPa on video understanding tasks. Specifically, we use the image encoder to process each frame separately and concatenate all resulting tokens. A LiT-Decoder [17] is then tuned on a mixture of six video understanding tasks. Despite no video has been seen during pretraining, LocCa showcases the video understanding capabilities and outperforms CapPa on most of the datasets. In particular, for the video captioning task on VATEX [96] dataset, LocCa achieves a CIDER score of 65.0 vs 64.0 of CapPa. For the video QA task on MSVD [97] dataset, LocCa obtains an accuracy of 50.9 vs 50.0 of CapPa. More details could be found from Appendix A.8.

### 4.3  Qualitative Results

In this section, we discuss the raw LocCa model's zero-shot capability to detect multiple objects using its own decoder, despite only a single object per example being observed during pretraining. To achieve this, we employ beam search when decoding the output from LocCa. This involves using the prompt of a single task prefix "*GCap:*" to instruct LocCa to predict the RoIs along with their labels. As shown in Appendix Fig. 6, we observe that depending on the setting we get lower number of bounding boxes with correct class names, or higher number of boxes with class names which start to contain noise. We provide more discussions in Appendix B. Nevertheless, pre-trained LocCa model shows powerful capability after a short finetuning on a clean downstream dataset as shown in Table 1. Finding a decoding strategy which can output high number of boxes and quality labels at the same time is an open problem.

Table 6: Ablation study on applying loss on AREF and GCAP tasks during training.

| AREF | GCAP | INet | COCO | VQAv2 | GQA | RefCOCO | | | RefCOCO+ | | |
|------|------|------|------|-------|-----|------|-------|-------|------|-------|-------|
| | | | | | | val | testA | testB | val | testA | testB |
| ✓ | ✓ | 80.4 | 117.7 | 68.4 | 59.0 | 88.0 | 90.8 | 84.0 | 80.2 | 84.7 | 73.8 |
| ✓ | ✗ | 79.7 | 115.9 | 67.5 | 57.9 | 86.8 | 89.6 | 83.0 | 77.7 | 83.5 | 71.2 |
| ✗ | ✓ | 79.7 | 114.8 | 67.4 | 57.7 | 83.7 | 87.2 | 80.7 | 72.3 | 77.9 | 66.6 |
| ✗ | ✗ | 78.3 | 111.0 | 66.2 | 54.3 | 75.2 | 78.8 | 69.5 | 64.7 | 71.0 | 55.9 |

## 4.4 Ablations

**Pretraining and transfer resolutions**  We present results with different pretrain resolution and transfer resolution combinations in Figure 3 (a). To get a LocCa model with higher 384 pretrain resolution, we finetune the 224 resolution model using the target 384 resolution for 900M examples seen on the same dataset. In Figure 3 (a), we transfer both the 224 resolution model and the 384 resolution model to RefCOCOs using different transfer resolutions (224, 384, 640) as marked on the x-axis. We find that the higher pretrain model achieves slightly higher or competitive results compared to the 224 resolution models. We opt for the 224 resolution pretrain models by default in this paper for simplicity. More results with high pretrain resolution could be found in the Appendix A.10.

**Coordinate tokenization**  Previous studies [18, 51] often tokenize object coordinates by adding a location vocabulary. In contrast, LocCa simplifies this process by directly converting integral coordinates into textual strings, which are then tokenized with the same text tokenizer. This section presents an ablation study on the RefCOCO benchmarks to examine the differences between these two options. As shown in Figure 3 (b), both tokenization strategies for box coordinates yield remarkably similar results. However, our approach is notably simpler and more straightforward.

**Selection of pretraining tasks**  To evaluate the importance of the selected tasks for pretraining LocCa, an ablation study was conducted focusing on the effects of AREF and GCAP tasks. Specifically, the study explored the impact on LocCa by removing these tasks individually and collectively, with the model defaulting to the CapPa baseline when both are excluded. All these models are pretrained on the WebLI split for 900M examples with the resolution of $224^2$px, and subsequently evaluated on the holistic tasks using a LiT-Decoder setup and on RefCOCOs by fine-tuning the whole model. As shown in Table 6, incorporating any location-aware task into the pretraining of LocCa yields significant performance improvements, particularly evident in the RefCOCO results. Interestingly, incorporating solely the GCAP task leads to a marked improvement in RefCOCO performance compared to the CapPa baseline, despite GCAP not being directly aligned with the AREF task. This highlights the importance of introducing object concepts for enhancing regional-level understanding, which is beneficial for fine-grained visual comprehension and applicable to other object-sensitive tasks. Furthermore, the combined inclusion of both AREF and GCAP tasks yields even better results, demonstrating their complementary nature in improving LocCa's performance.

## 5  Conclusion

LocCa introduces a novel visual pretraining paradigm, using natural language interface to construct the proxy location-aware pretrain tasks. This model excels in understanding both the overall scene and specific spatial details, setting new performance standards on location-aware tasks while preserving the capability on holistic image understanding. LocCa simplifies the process of blending location information with visual data, offering significant improvements on tasks requiring detailed spatial awareness. Our contributions pave the way for advanced model capabilities in processing a broad spectrum of vision-language tasks, demonstrating LocCa's versatility and effectiveness.

LocCa already demonstrates superior zero-shot detection capability in qualitative results. However, it lacks the capability for zero-shot segmentation due to the absence of pretraining on pixel-level annotations, and we leave this for future work. Building on the robust foundation of LocCa, a promising direction for future work involves enhancing its pixel-level precision through the incorporation of segmentation tasks during the pretraining phase. This extension aims to equip LocCa with a more nuanced understanding of images, enabling it to discern and interpret intricate details and textures with unparalleled accuracy, further broadening its applicability across diverse vision-language tasks.

## Acknowledgements

We thank Matthias Minderer and Jeremiah Harmsen for their valuable feedback on the manuscript, and Debidatta Dwibedi and Xingyi Zhou for their feedback on dense captioning evaluators. We also thank Paul Voigtlaender for discussions on the referring expression segmentation task. Finally, we thank the Google DeepMind team for providing a supportive research environment. We used the `big_vision` codebase [98, 99] for all experiments in this project.

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

# A   Details on transfer results and ablations

## A.1   Transfer to downstream tasks

In this study, we introduce three methodologies to evaluate the performance of the pretrained LocCa across a variety of downstream tasks. The first methodology focuses on assessing the efficacy and generalization capability of the standalone pretrained visual encoder. In this setup, the visual encoder is kept frozen, and we employ two main strategies: (i) Drawing inspiration from Lit-Decoder [17], we train a multi-task decoder from scratch for all downstream tasks. This approach is particularly suited for holistic image understanding and video-related tasks due to the broad range of tasks, allowing for a simplified evaluation process with a single decoder; and (ii) Aligning with the recent advancements [10, 47, 31, 52] that showcase the exceptional results achieved by integrating a pretrained visual encoder with a large language model [100] across a multitude of downstream tasks, we adopt the training strategy outlined in PaLI-3 [31], substituting their visual encoder with our LocCa pretrained encoder.

The second methodology entails assessing the performance of the full LocCa model by fine-tuning it on downstream tasks using a minimal learning rate and weight decay. Given the high cost associated with fine-tuning the entire model, we selectively evaluate its adaptability on just two tasks: image captioning for holistic image understanding and referring expression comprehension for location-aware understanding.

The third methodology involves combining the pretrained LocCa visual encoder with a task-specific pretrained decoder for selected tasks, such as object detection. This approach is designed to leverage the strengths of LocCa's visual encoder in particular scenarios, notably in contexts requiring fine-grained object-centric understanding.

## A.2   LiT-Decoder setup for Referring Expression Comprehension and holistic image understanding tasks

For the evaluation of different frozen vision backbones on classification, captioning, and VQA with the protocol from [17] (LiT-Decoder) we rely on the hyper-parameters from [14] (which only differ from [17] in the data mixing strategy).

We also apply this protocol to Referring Expression Comprehension to compare the encoders only (without pretrained decoders), with small modifications. Specifically, besides training on a data mix of the different RefCOCO variants, we reduce the number of decoder layers from 12 to 6, and reduce the learning rate from $10^{-3}$ to $3 \times 10^{-4}$. Also note that transfer is done at the pretraining resolution of $224^2$px, unlike the fine-tuning transfers which are done at higher resolution.

## A.3   Referring Expression Segmentation

For referring expression segmentation, we extend the REC task by adding a suffix "*Mask*:" to the text, followed by 16 integer numbers corresponding to segmentation tokens. The segmentation tokens specify the precise shape of the segmentation task within the bounding box that is predicted as part of the REC task. We use the vector-quantized variational auto-encoder (VQ-VAE) from [86], which was trained as in [31] on OpenImages data [101]. The VQ-VAE converts a single channel $64 \times 64$ pixel mask into a sequence of 16 integer values from a discrete codebook of 128 tokens. LocCa is trained to predict the 16 segmentation tokens, and the VQ-VAE decoder is then used to convert these tokens to a $64 \times 64$ pixels segmentation mask. This mask is then resized to the predicted box coordinates (with bilinear interpolation), before computing the IoU with the ground truth mask.

In Table 7, we showcase the results of employing the full encoder-decoder LocCa model and fine-tuning it for RES task. It achieves competitive results even compared to the state-of-the-art PaLI-3 model, with considerably smaller model size.

## A.4   Vision-Language Models with LocCa

We follow the setup of PaLI-3 [31] ablations for all PaLI-3 transfer evaluations. More specifically, we pretrain PaLI-3 stage 1 (frozen image encoder, $224^2$px resolution) with batch 16k for 13k steps (i.e. 208 million examples). Then we fine-tune PaLI-3 model on each downstream task separately,

Table 7: mIoU on RefCOCO **segmentation** results when fine-tuning the full LocCa model, and comparison with models from the literature.

| MODEL | RefCOCO | | | RefCOCO+ | | | RefCOCOg | |
|---|---|---|---|---|---|---|---|---|
| | val | testA | testB | val | testA | testB | val-u | test-u |
| PaLI-3[31] | 77.33 | – | – | 72.53 | – | – | 72.72 | – |
| PolyFormer$_L$[102] | 76.94 | 78.49 | 74.83 | 72.15 | 75.71 | 66.73 | 71.15 | 71.17 |
| LocCa$_L$ | 76.98 | 78.25 | 72.90 | 71.25 | 76.52 | 63.67 | 69.51 | 70.44 |

Table 8: Details of PaLI-3 [31] transfer evaluations, where the model is fine-tuned for each task separately.

| PARAM | COCO | VQAv2 | OKVQA | TextVQA | ST-VQA | TallyQA |
|---|---|---|---|---|---|---|
| steps (k) | 20 | 10 | 5 | 10 | 10 | 10 |
| batch size | 256 | 256 | 128 | 128 | 128 | 128 |
| learning rate | 1e-4 | 1e-4 | 3e-5 | 1e-4 | 1e-4 | 3e-5 |
| weight decay | 1e-4 | 5e-5 | 1e-4 | 1e-4 | 1e-4 | 1e-4 |

keeping image encoder frozen and the same $224^2$ resolution. The details of each task is listed in Table 8.

## A.5 Object Detection

To validate the effectiveness of LocCa compared with image-text pretrained baselines on object detection, we employ a ViT-B/16 encoder and pretrain all these models on our WebLI subset for 9B examples with $224^2$px. Subsequently, we integrate the pretrained visual encoder with a 6-layer Objects365 pretrained decoder, adapting the combined model for the COCO detection task. This adaptation processes inputs at a $640^2$px and is designed to predict up to 25 bounding boxes, using a batch size of 256. First we fine-tune the model for 10k steps with learning rate of $10^{-4}$, employing a standard auto-regressive loss. This is followed by an additional fine-tuning phase focused on maximizing the mAP reward for 20k steps with learning rate of $10^{-6}$.

## A.6 Zero-shot transfer

Following locked-image tuning [36], we freeze the pre-trained vision encoder and train a text encoder contrastively to perform zero-shot classification and zero-shot retrieval on downstream tasks. We take the L/14 vision encoder from a LocCa$_L$ model, and then train a text encoder from scratch to pair with the frozen L/14 model. Input image is resized to resolution $224^2$px. The model is trained for 3B seen examples, with a standard $10^{-3}$ learning rate and $10^{-4}$ weight decay. Beyond that, we also attach a randomly initiated MAP head [4] to the L/14 vision encoder and finetune the MAP head following the CapPa baseline [14].

## A.7 Semantic segmentation on ADE20k

We follow the setup of Segmenter [94] which fine-tunes the vision encoder jointly with a linear head predicting the semantic label for every patch, followed by upsampling. The input image resolution is set to $512 \times 512$, and we apply random resizing with aspect ratio in the range $[0.5, 2.0]$, photometric jitter, and random horizontal flipping. We train for 160k steps with batch 16 using Adam with learning rate $3 \times 10^{-5}$ and decoupled weight decay of 0.01. To accommodate variable inference resolution for evaluation, we apply our model in sliding-window fashion at the training resolution.

## A.8 Experiments on video understanding

Even though LocCa has never seen any videos during pretraining, its training recipe results in an improvement upon CapPa in many video-related tasks, including captioning, QA, and classification. Here, we follow the setup used in PaLI-3 [31], in which we use the image encoder to process each

Table 9: Video evaluation results with LocCa-pretrained vision encoder. We use CIDEr [107] (C) and BLEU-4 (B) for captioning, matching accuracy (A) for QA and YTBB, and the Jackard index (J) for VLOGS. LocCa improves upon CapPa, particularly when using the larger ViT-L/16 encoder.

| MODEL | Captioning | | | | QA | | Classification | |
|---|---|---|---|---|---|---|---|---|
| | MSRVTT | | VATEX | | MSRVTT-QA | MSVD-QA | YTBB | VLOGS |
| | C | B | C | B | A | A | A | J |
| CapPa/B | $47.4_{\pm.3}$ | $41.8_{\pm.3}$ | $49.7_{\pm.2}$ | $28.9_{\pm.0}$ | $37.7_{\pm.1}$ | $45.6_{\pm.9}$ | $94.6_{\pm.1}$ | $52.7_{\pm.2}$ |
| LocCa/B | $47.6_{\pm.4}$ | $41.8_{\pm.0}$ | $49.7_{\pm.2}$ | $29.1_{\pm.0}$ | $37.9_{\pm.1}$ | $44.3_{\pm.4}$ | $94.2_{\pm.2}$ | $\mathbf{54.0_{\pm.3}}$ |
| CapPa/L | $55.6_{\pm.3}$ | $47.1_{\pm.1}$ | $64.0_{\pm.2}$ | $36.2_{\pm.1}$ | $39.9_{\pm.0}$ | $50.0_{\pm.3}$ | $94.8_{\pm.1}$ | $58.0_{\pm.3}$ |
| LocCa/L | $55.1_{\pm.2}$ | $47.3_{\pm.4}$ | $\mathbf{65.0_{\pm.4}}$ | $\mathbf{36.8_{\pm.1}}$ | $\mathbf{40.5_{\pm.2}}$ | $\mathbf{50.9_{\pm.5}}$ | $94.7_{\pm.0}$ | $58.6_{\pm.1}$ |

frame separately and concatenate all resulting tokens. Then, we tune a LiT decoder [17] by freezing the pretrained image encoder while training a two-layer text decoder from scratch with resolution 224 on a mixture of six video tasks, each of which is given an equal weight during training; i.e. the same number of examples are seen from each dataset. We use C4 tokenizer [65]. The model is trained for 5K steps, using a maximum text length of 64 tokens and batch size 256. We use label smoothing of 0.1 [103], learning rate $10^{-3}$ with weight decay $10^{-4}$, and cosine learning rate schedule with 10% warm-up. Encoder ViT-B/16 is pretrained on 3B examples while ViT-L/16 is pretrained on 9B examples.

The six datasets are: (1) **MSRVTT** [104], a collection of video clips annotated with 20 captions, (2) **VATEX** [96], another collection with 10 captions each, (3) **MSRVTT-QA** [97], which contains QA pairs generated from video descriptions using an automated tool, (4) **MSVD-QA** [97], another collection of QA pairs, (5) **YTBB** [105], a video classification dataset containing 23 classes, and (6) **VLOGS** [106], a multi-label classification containing 23 classes. Of the six datasets, VLOGS is location-related, since the task involves tracking the movement of a hand. Indeed, while LocCA leads to favorable improvements overall over CapPa, especially using ViT-L/16 encoder, the gain is particularly notable in VLOGS as expected. Table 9 summarizes all the results.

**Grounded captioning**

For the grounded captioning task, we follow the standard setup that generates regional captions based on a given bounding box location [58, 108, 109]. We report the grounded captioning results on the Visual Genome dataset and compare them with state-of-the-art methods.

As shown in Table 10, LocCa (0.6B, without LLM), outperforms GPT4RoI [108] (7B, with LLM) on both METEOR and CIDEr scores. METEOR evaluates the precision, recall, and alignment of words between the generated and reference captions, while CIDEr assesses the similarity of n-grams between them. When compared with GLaMM [109] (7B, with LLM), LocCa performs better on METEOR but lags on CIDEr. This difference can be attributed to the relatively simple captions in Visual Genome, which typically consist of only a few words. LocCa uses a simpler decoder (0.3B params only) that closely matches the reference captions in terms of word choice and order, which aligns well with the dataset's simple nature. GLaMM, on the other hand, uses a more complex LLM as its decoder, which likely generates more diverse captions that include n-grams that match the reference captions more effectively.

It is important to note that the pretrained LocCa encoder complements multimodal LLMs (i.e. as a better alternative option to the CLIP encoder), and we anticipate further performance gains on downstream tasks when combining both.

Table 10: Grounded captioning results on the Visual Genome dataset.

| Model | # Param | mAP | METEOR | CIDEr |
|---|---|---|---|---|
| GRIT | 1B | 15.5 | 17.1 | 142 |
| GPT4RoI | 7B | - | 17.4 | 145.2 |
| GLaMM | 7B | - | 19.7 | **180.5** |
| LocCa$_L$ | 0.6B | **34.5** | **20.7** | 157.0 |

### A.9 Experiments on grounded captioning

### A.10 Impact of high resolution pretraining and tokenization strategies

In this section, we provides the complete results of the ablation studies on different image resolution pretraining in Fig. 4, and different tokenization strategies for box coordinates in Fig. 5, focusing on the RefCOCO benchmarks.

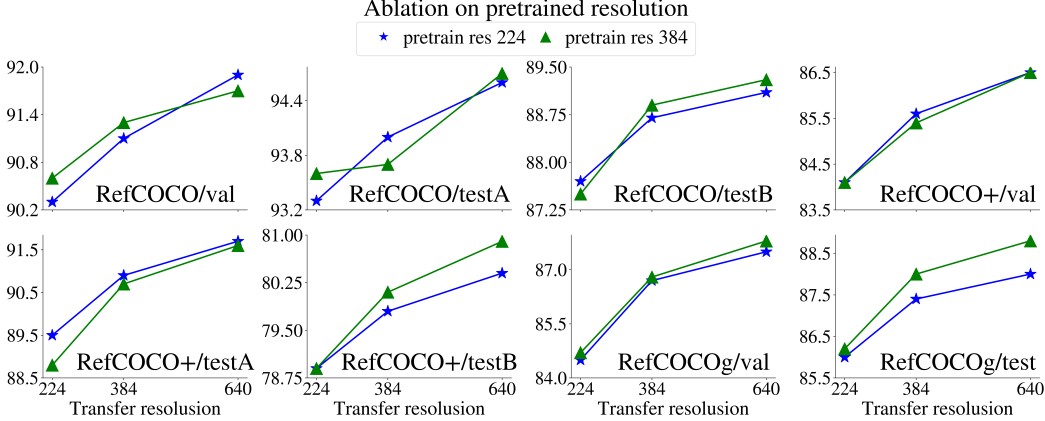

Figure 4: Ablation studies on string vs special tokenization for box coordinates. The image resolution is 224.

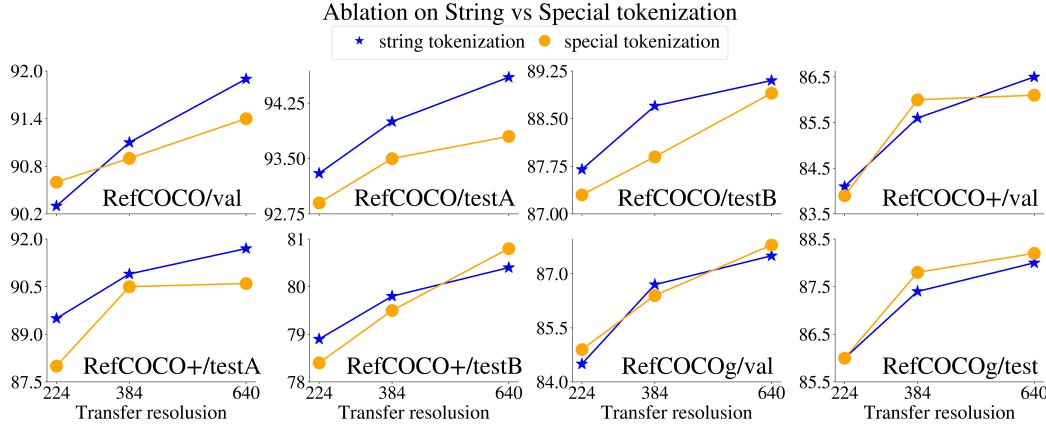

Figure 5: Ablation studies on impact of different pretrained image resolutions on string token, we use string tokenization for box coordinates.

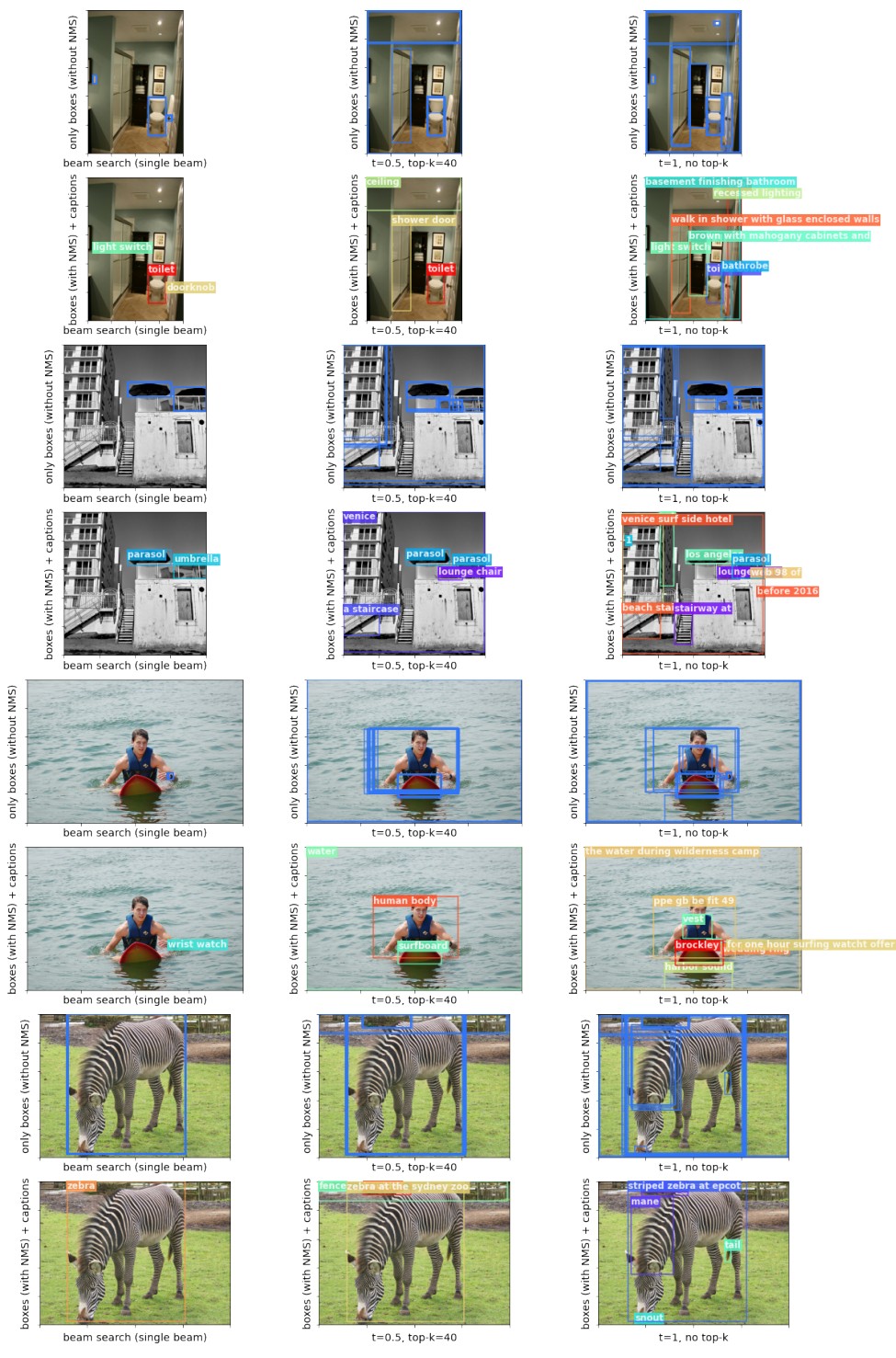

Figure 6: Visualizations of LocCa$_L$'s zero-shot predictions on Visual Genome[57]: from left to right we increase noise when sampling. The top rows show all the boxes without non-maximum suppression and the bottom rows show captions for boxes passing the non-maximum suppression filter. We can see that adding more noise increases the variety and amount of decoded boxes and at the same time degrades the quality of captions.

## B  More Visualization Results

We provide visualizations of LocCa$_L$'s zero-shot predictions on Visual Genome[57] in Fig. 6. We have two notable observations: (i) When comparing the top rows to the bottom rows, LocCa excels at identifying foreground regions, yet the box regions exhibit significant overlap without non-maximum suppression. This is because only one single object per example is observed during LocCa pretraining, leading to the selection of RoIs that are highly random. (ii) From left to right we increase noise during sampling, which increases the variety and quantity of decoded boxes, while simultaneously degrades the quality of captions. We hypothesize that box sampling demands higher noise levels to explore to explore more diverse RoIs, whereas text sampling requires lower noise levels to ensure the generated texts are highly semantic relevant.

## C  Duplicate image ids for RefCOCOs

Recent studies in referring expression comprehension [52, 76, 77, 18] typically train their models using the combined training sets of RefCOCO, RefCOCO+, and RefCOCOg. However, the splits of these three datasets largely overlap, implying that methods trained on more than half of the test images. In Table 11, we present the image ratio of validation and test splits of RefCOCOs that overlap with the combined training sets. Adhering to the general principle, we provide a list of image IDs in the training set of RefCOCO benchmarks that are duplicated with validation and test images.

Table 11: The image ratio of validation and test splits of RefCOCOs that overlap with the combined training sets.

|       | RefCOCO | | | RefCOCO+ | | | RefCOCOg | |
|-------|------|-------|-------|------|-------|-------|-------|--------|
|       | val  | testA | testB | val  | testA | testB | val-u | test-u |
| Ratio | 61.2% | 60.5% | 65.1% | 61.2% | 60.5% | 65.1% | 48.8% | 48.3% |

154, 309, 605, 656, 716, 797, 839, 909, 977, 1298, 1488, 1507, 1947, 1958, 1994, 2083, 2342, 2400, 2411, 2448, 2567, 2742, 2843, 2964, 3000, 3178, 3293, 3518, 3751, 4244, 4424, 4477, 4587, 5152, 5377, 5424, 5434, 5508, 5614, 5632, 5862, 5962, 6026, 6051, 6407, 6747, 6842, 6943, 6964, 7028, 7145, 7277, 7393, 7476, 7504, 7601, 7621, 7944, 7945, 7946, 8063, 8300, 8320, 8429, 8936, 9017, 9018, 9029, 9057, 9218, 9723, 9822, 10094, 10179, 10229, 10471, 10710, 10948, 11065, 11282, 11324, 11774, 12224, 12377, 12382, 12440, 12495, 12614, 12790, 13352, 13576, 13670, 13763, 13856, 14008, 14025, 14160, 14283, 14468, 14484, 14502, 14864, 15190, 15195, 15554, 15809, 16089, 16273, 16465, 16616, 16659, 16669, 16725, 16796, 16814, 16836, 16870, 17236, 17468, 17566, 17587, 17938, 17997, 18075, 18089, 18093, 18211, 18244, 18370, 18542, 18780, 19123, 19374, 19501, 19789, 19959, 19967, 20044, 20156, 20188, 20279, 20513, 20619, 20917, 21292, 21504, 21750, 21830, 22102, 22195, 22222, 22287, 22575, 22740, 22928, 23014, 23141, 23194, 23420, 23538, 23967, 24038, 24086, 24129, 24404, 24689, 24706, 24762, 24842, 24939, 25237, 25353, 25414, 25515, 25628, 26052, 26421, 26438, 26583, 26997, 27070, 27149, 27424, 27495, 27750, 27763, 28012, 28038, 28069, 28154, 28281, 28451, 28560, 28824, 28974, 28988, 29304, 29456, 29473, 29601, 29752, 29799, 30203, 30274, 30340, 30418, 30631, 30973, 31112, 31187, 31230, 31329, 31374, 31382, 31418, 31838, 32061, 32105, 32289, 33017, 33527, 33572, 33991, 33992, 34223, 34616, 34739, 34810, 35045, 35132, 35230, 35265, 35322, 35473, 35558, 35571, 35796, 36017, 36445, 36488, 36546, 36755, 36981, 37089, 37282, 37539, 37582, 37698, 37719, 37800, 37847, 37862, 38033, 38266, 38552, 38890, 39159, 39185, 39258, 39629, 39802, 39812, 40130, 40346, 40735, 41713, 41730, 41818, 41988, 42696, 42804, 43163, 43609, 43655, 43664, 43813, 43892, 44123, 44266, 44298, 44637, 44901, 44960, 45226, 45464, 45659, 45672, 45840, 46055, 46385, 46519, 46609, 46809, 47093, 47175, 47198, 47294, 47347, 47357, 47391, 47451, 47554, 47774, 47928, 48267, 48432, 48572, 48665, 48707, 48937, 49022, 49073, 49866, 50105, 50134, 50161, 50601, 50961, 51001, 51052, 51550, 51563, 51706, 51835, 51965, 52109, 52168, 52219, 52448, 52484, 52626, 52729, 52751, 52929, 53004, 53232, 53304, 53335, 53370, 53388, 53601, 53928, 53929, 54402, 54541, 54572, 54764, 54805, 54806, 55226, 55232, 55385, 55733, 55764, 55873, 56028, 56032, 56604, 56616, 56632, 56667, 56699, 56738, 57551, 57689, 57699, 57828, 57870, 58184, 58403, 58633, 58836, 59079, 59231, 59382, 59483, 59556, 59593, 59654, 59816, 59947, 60043, 60155, 60182, 60639, 61159, 61328, 61372, 61459, 61460, 61842, 61843, 61936, 62038, 62057, 62131, 62203, 62233, 62263, 62455, 62477, 62759, 63084, 63238, 63334, 63337, 63347, 63754, 63820, 63867, 64317, 64392, 64962, 65011, 65136, 65769, 65841, 65842, 66236, 66518, 66566, 66593, 66637, 66737, 67356, 67438, 67615, 67748, 68139, 68397, 68430, 68459, 68786, 69047, 69344, 69432, 69488, 69971, 69978, 70000, 70094, 70161, 70415, 70718, 70745, 70755, 71099, 71229, 71271, 71399, 71714, 71970, 71989, 72111, 72396, 72565, 72592, 72629, 72731, 72947, 72995, 73174, 73583, 73591, 73602, 73610, 73671, 73680, 73999, 74127, 74156, 74201, 74215, 74577, 74663, 74925, 74942, 74945, 74996, 75590, 75691, 75843, 75881, 75924, 76245, 76414, 76590, 76781, 76882, 76885, 77005, 77174, 77332, 77377, 77380, 77417, 78009, 78221, 78274, 78425, 78482, 78536, 78572, 79083, 79313, 79441, 79456, 79611, 79701, 79783, 79887, 80207, 80305, 80472, 80521, 80634, 80782, 80818, 80835, 81065, 81128, 81200, 81283, 81372, 81768, 81799, 81810, 82083, 82228, 82484, 83005, 83353, 83448, 83725, 83815, 83866, 84162, 84167, 84259, 84558, 84594, 84712, 84800, 84803, 85549, 85893, 85960, 86075, 86216, 86217, 86459, 86549, 86654, 86750, 86754, 86869, 87214, 87235, 87458, 87522, 87569, 87671, 87737, 87792, 88200, 88609, 88653, 88671, 88726, 88868, 89005, 89181, 89748, 89651, 89734, 89788, 89882, 89921, 89931, 90277, 90310, 90350, 90444, 90569, 90573, 90830, 90985, 91055, 91056, 91130, 91288, 91784, 92009, 92165, 92439, 92480, 92646, 92685, 92694, 93171, 93531, 93581, 93786, 93885, 94459, 94618, 94826, 95061, 95257, 95518, 95562, 95676, 95812, 96177, 96244, 96338, 96475, 96728, 96808, 96859, 96958, 97411, 97563, 97795, 97818, 98447, 99040, 99159, 99211, 99293, 99451, 99727, 100182, 100209, 100485, 100586, 100707, 101479, 101503, 101615, 101697, 101807, 101832, 101882, 101891, 102144, 102208, 102281, 102290, 102662, 102667, 103223, 103251, 103419, 103510, 104126, 104248, 104277, 104304, 104344, 104410, 104692, 104752, 105026, 105063, 105219, 105470, 105660, 105714, 105719, 105859, 106100, 106148, 106315, 106383, 106557, 106832, 106978, 106994, 107009, 107100, 107156, 107767, 107846, 108375, 108499, 108920, 109088, 109553, 109654, 109778, 109986, 110230, 110252, 110447, 110989, 111040, 111045, 111195, 111543, 111705, 111754, 111842, 111992, 112040, 112226, 112495, 112577, 112707, 113032, 113123, 113152, 113244, 113676, 113844, 113998, 114060, 114326, 114459, 114786, 114801, 114807, 115505, 115524, 115564, 116040, 116049, 116603, 116607, 116824, 116832, 116882, 117117, 117182, 117447, 117578, 117677, 117772, 117969, 118169, 118277, 118413, 118543, 118697, 118780, 118827, 119093, 119129, 119263, 119534, 119714, 119974, 120155, 120274, 120333, 120431, 120836, 121445, 121453, 121575, 121619, 121683, 121903, 121938, 121965, 121994, 122099, 122259, 122436, 122459, 122916, 122918, 123180, 123366, 124030, 124055, 124069, 124169, 124178, 124694, 124711, 124751, 124786, 124804, 125658, 125690, 125774, 125785, 125882, 126381, 126447, 126534, 126737, 126825, 126909, 127282, 127316, 127388, 127543, 127560, 127615, 127629, 127729, 127945, 128106, 128127, 128136, 128282, 128385, 128434, 128475, 128599, 128647, 128775, 128938, 129551, 129771, 129806, 130116, 130163, 130215, 130324, 130339, 130518, 130872, 131058, 131074, 131127, 131277, 131449, 131587, 131595, 131816, 132183, 132529, 132574, 132585, 132617, 132746, 133295, 133384, 133609, 133654, 133905, 133940, 134100, 134176, 134420, 134447, 134474, 134755, 134799, 135332, 135482, 135539, 135694, 135815, 135822, 136092, 136230, 136232, 136310, 136563, 136736, 136861, 137052, 137378, 137516, 137715, 137724, 137770, 137918, 138131, 138436, 138507, 138567, 138910, 139068, 139173, 139324, 139359, 139429, 139568, 139696, 139728, 139763, 139775, 139811, 139914, 140053, 140108, 140291, 140733, 140738, 140850, 140954, 141101, 141603, 141682, 141702, 141827, 141952, 142322, 142431, 142439, 142969, 143003, 143258, 143323, 143334, 143470, 143665, 144272, 144275, 144320, 144495, 144519, 144574, 144817, 144832, 144851, 144996, 145178, 145192, 146561, 147466, 147701, 147710, 147733, 147753, 147760, 147838, 148044, 148047, 148809, 148937, 149180, 149202, 149253, 149498, 149556, 149566, 149616, 149916, 150044, 150100, 150477, 150614, 150704, 151163, 151178, 151236, 151265, 151523, 151699, 151756, 151908, 152003, 152197, 152237, 152238, 152273, 152501, 152556, 152871, 152954, 153340, 153475, 153591, 153671, 153749, 153814, 153845, 154244, 154713, 154895, 154911, 155107, 155268, 155269, 155379, 155549, 155860, 155904, 156125, 156258, 156608, 156757, 156823, 156827, 156914, 156939, 157125, 157194, 157242, 157344, 157424, 157714, 157744, 157834, 157926, 158051, 158201, 158362, 158686, 158701, 159038, 159682, 159768, 159957, 160101, 160291, 160503, 160614, 160648, 160688, 160852, 161719, 161818, 161865, 162046, 162102, 162283, 162300, 162551, 162645, 162760, 163054, 163089, 163266, 163394, 163559, 163991, 164043, 164100, 164381, 164855, 164935, 165077, 165199, 165555, 165868, 166073, 166230, 166328, 166653, 166762, 167169, 167264, 167755, 167765, 168022, 168179, 168366, 168482, 168643, 168865, 169179, 169529, 169656, 169725, 170327, 170398, 170483, 170623, 170636, 170683, 170689, 170712, 170809, 170976, 170980, 171262, 171478, 171536, 171729, 171943, 172669, 172957, 173032, 173056, 173484, 173538, 173550, 173814, 173925, 174059, 174137, 174554, 174700, 174749, 174774, 174876, 174892, 175112, 175116, 175118, 175162, 175195, 175284, 175405, 175480, 175523, 175881, 176008, 176032, 176229, 176342, 176386, 176790, 176810, 176871, 177019, 177193, 177289, 177314, 177353, 177472, 177516, 177817, 177821, 177915, 177917, 178131, 178192, 178492, 178763, 178874, 178987, 179011, 179164, 179390, 179618, 179763, 179969, 180179, 180220, 180285, 180354, 180559, 180578, 180667, 181054, 181176, 181475, 181681, 181929, 182406, 182505, 182571, 182642, 182863, 183022, 183236, 183237, 183392, 183435, 183538, 183626, 183646, 183653, 183788, 183835, 183923, 184106, 184184, 184513, 185153, 185258, 186131, 186198, 186246, 186255, 186336, 186605, 187147, 187511, 187939, 188087, 188184, 188587, 188621, 188845, 188911, 189330, 189353, 189646, 189924, 190026, 190216, 190219, 190277, 190513, 190617, 190732, 191068, 191305, 191327, 191561, 191754, 192319, 192407, 192476, 192524, 192878, 193042, 193168, 193171, 193682, 193829, 194056, 194154, 194193, 194438, 194448, 194550, 194669, 194677, 194679, 194726, 194758, 194847, 195027, 195188, 195525, 195861, 196111, 196112, 196156, 196170, 196198, 196653, 196971, 197222, 197251, 197323, 197401, 197525, 197591, 197663, 198277, 198651, 198785, 199234, 199331, 199485, 199835, 199836, 199888, 199963, 200010, 200181, 200377, 200404, 201184, 201322, 201634, 201687, 202057, 202755, 203034, 203036, 203098, 203108, 203175, 203458, 203459, 203994, 204053, 204294, 204759, 204792, 204800, 205000, 205131, 205202, 205223, 205250, 205460, 205794, 205963, 206377, 206486, 206628, 206731, 206968, 207381, 207467, 207496, 207629, 208075, 208379, 208396, 208724, 208845, 208963, 209089, 209185, 209191, 209356, 209449, 209537, 209563, 209603, 209654, 209844, 209993, 210187, 210252, 210271, 210279, 210604, 210710, 210773, 210848, 211138, 211570, 211576, 211643, 212070, 212247, 212450, 212532, 212628, 212635, 212641, 212757, 212974, 213408, 213419, 213426, 214523, 214875, 215026, 215191, 215201, 215243, 215357, 215407, 215421, 215563, 215701, 215908, 216391, 216579, 216676, 216840, 217043, 217151, 217276, 217429, 217487, 217799, 217893, 217959, 217978, 218057, 218096, 218145, 218579, 218734, 218984, 219127, 219248, 219349, 219535, 219680, 219966, 220037, 220053, 220148, 220529, 221053, 221169, 221187, 221252, 221625, 221674, 221794, 221880, 221889, 222113, 222209, 222676, 222977, 223023, 223078, 223165, 223270, 223603, 223650, 223790, 223831, 223909, 224056, 224060, 224168, 224541, 224692, 224734, 224753, 224821, 224838, 225069, 225468, 225539, 225579, 225604, 225641, 225755, 226102, 226176, 226329, 226348, 226357, 226460, 226552, 226681, 226712, 226734, 226817, 226961, 226966, 227198, 227205, 227520, 227554, 228000, 228119, 228133, 229041, 229105, 229193, 229362, 229415, 229422, 229598, 229678, 229825, 230436, 230515, 230545, 230559, 230570, 230893, 231047, 231087, 231337, 231992, 232371, 232770, 233007, 233022, 233071, 233153, 233642, 233841, 233871, 233878, 234244, 234653, 234699, 234819, 235468, 235582, 235802, 236036, 236174, 236381, 236397, 236556, 236718, 236961, 237110, 237137, 237193, 237273, 237340, 237355, 237367, 237510, 237834, 237922, 237976, 238007, 238070, 238187, 238231, 238238, 238502, 238618, 238630, 238667, 238713, 239263, 239461, 239559, 239596, 239654, 239772, 239933, 240225, 240331, 240339, 240378, 240500, 240521, 240586, 240662, 240945, 241369, 241887, 242076, 242090, 242145, 242213, 242350, 242453, 242506, 242583, 242709, 242745, 242807, 242854, 243071, 243153, 243336, 243574, 243717, 243824, 243839, 243959, 244016, 244171, 244387, 244425, 244528, 244616, 244825, 244836, 244839, 244844, 244846, 244875, 244983, 245118, 245326, 245946, 246084, 246089, 246342, 246356, 246539, 246641, 246753, 246777, 246959, 247082, 247110, 247265, 247271, 247368, 247660, 247979, 248052, 248337, 248579, 248640, 248666, 248730, 248733, 248835, 248932, 248957, 248979, 249805, 250293, 250295, 250569, 251368, 251523, 251868, 252025, 252136, 252373, 252492, 252768, 252937, 253064, 253087, 253229, 253251, 253335, 253430, 253796, 253834, 254577, 254821, 255203, 256190, 256215, 256364, 256659, 256930, 256951, 257301, 257392, 257451, 257576, 257804, 257815, 257867, 257874, 258165, 258237, 258249, 258505, 258571, 258705, 259120, 259375, 259484, 259514, 259595, 259655, 259809, 260029, 260118, 260181, 260206, 260299, 260317, 260360, 260448, 260668, 260932, 260957, 261283, 261503, 261521, 261673, 261696, 261720, 261990, 262031, 262180, 262239, 262528, 263039, 263176, 263420, 263516, 263924, 264016, 264058, 264165, 264502, 264567, 264741, 264781, 264846, 264885, 265292, 265329, 265625, 265713, 265766, 265980, 266207, 266228, 266240, 266442, 266515, 266816, 266859, 266898, 267049, 267189, 267779, 267794, 267815, 267851, 267898, 267907, 268260, 268428, 268644, 268726, 268881, 268897, 269160, 269199, 269245, 269380, 269504, 269605, 269890, 270111, 270391, 270696, 270844, 271106, 271447, 271641, 272022, 272058, 272235, 272255, 272299, 272310, 272670, 272716, 272729, 272773, 273197, 274139, 274275, 274499, 274642, 274667, 274770, 274786, 274839, 274853, 274986, 275544, 275556, 275658, 275707, 275709, 275775, 275811, 275917, 275932, 276089, 276244, 276354, 276417, 276621, 276666, 276686, 276711, 276740, 276845, 276874, 277202, 277267, 277418, 277439, 277507, 278461, 278931, 279076, 279377, 279485, 279530, 279762, 279882, 280018, 280156, 280191, 280257, 281051, 281237, 281464, 281790, 282067, 282142, 282310, 282514, 282568, 283263, 283479, 283573, 283666, 283673, 283937, 284348, 284639, 284778, 284934, 284964, 285000, 285064, 285093, 285170, 285214, 285307, 285395, 285529, 285548, 286000, 286051, 286132, 286190, 286359, 286411, 286469, 286745, 287140, 287249, 287302, 287567, 287718, 288559, 288610, 288943, 289140, 289282, 289696, 289782, 289791, 289866, 290072, 290098, 290114, 290185, 290265, 290354, 290370, 290549, 290620, 290938, 291039, 291366, 291526, 291897, 292116, 292271, 292386, 292498, 292558, 292751, 293293, 293311,

293489, 293853, 293966, 294409, 295370, 295578, 295759, 295940, 296093, 296191, 296267, 296360, 296474, 296614, 296635, 296747, 296760, 296894, 296984,
297011, 297019, 297251, 297266, 297360, 297363, 297527, 297665, 297764, 297984, 297997, 298110, 298160, 298312, 298360, 298639, 298793, 298931, 298956,
298983, 299029, 299041, 299051, 299085, 299122, 299123, 299463, 299594, 299859, 299933, 299959, 300021, 300197, 300239, 300624, 301109, 301158, 301218,
301413, 301461, 301943, 301970, 301988, 302199, 302353, 302397, 302415, 302885, 303144, 303247, 303360, 303541, 303923, 304092, 304125, 304319, 304406,
304603, 304833, 305105, 305106, 305141, 305219, 305231, 305267, 305492, 305546, 305550, 305564, 306275, 306393, 306483, 306837, 306967, 307082,
307190, 307242, 307322, 307475, 307671, 307757, 307881, 307968, 308089, 308139, 308210, 308265, 308524, 309034, 309084, 309135, 309280, 309400, 309409,
309706, 310006, 310158, 310360, 310457, 310518, 310759, 310780, 310851, 310865, 310897, 311273, 311388, 311620, 311709, 311773, 311890, 311933, 312154,
312205, 312247, 312282, 312390, 312454, 312608, 312748, 312785, 312886, 312924, 313071, 313073, 313164, 313206, 313209, 313360, 313518, 313569, 313724,
313786, 313873, 313946, 313950, 314065, 314237, 314254, 314319, 314462, 314734, 314920, 314951, 315411, 315555, 315581, 315751, 315944, 315961, 316170,
316293, 316446, 316671, 316801, 316971, 317054, 317349, 317391, 317659, 317805, 317905, 318117, 318183, 318203, 318528, 318953, 319062, 319192, 319396,
319543, 319644, 319685, 319735, 319866, 320059, 320077, 320125, 320137, 320292, 320308, 320371, 320390, 320403, 320611, 320667, 320721, 320834, 320957,
321305, 321673, 321766, 321960, 321969, 322090, 322411, 322630, 322634, 322698, 322726, 323052, 323147, 323149, 323213, 323240, 323389, 323664, 323722,
323728, 323734, 323960, 324528, 324677, 324732, 324910, 325229, 325302, 325362, 325494, 325548, 325950, 326092, 326237, 326357, 326475, 326836, 326841,
326966, 327132, 327198, 327209, 327258, 327338, 327404, 327561, 327843, 327998, 328113, 328214, 328298, 328663, 328855, 328917, 329058,
329141, 329343, 329501, 329502, 329528, 329616, 329724, 329963, 329993, 330040, 330094, 330223, 330284, 330342, 330671, 330716, 330752, 330785, 330991,
331216, 331222, 331326, 331331, 331419, 331505, 331544, 332133, 332385, 332905, 332976, 333207, 333225, 333383, 333498, 333748, 333842, 333922, 334139,
334529, 334714, 334742, 334775, 335066, 335076, 335107, 335304, 335362, 335376, 335524, 335525, 335697, 335752, 335758, 335865, 336242, 336267, 336350,
336406, 336503, 336683, 337147, 337156, 337164, 337255, 337445, 337509, 337628, 337689, 337691, 337704, 337808, 337976, 338025, 338214, 338218, 338385,
338819, 338872, 338978, 339051, 339283, 339453, 339579, 339597, 339816, 340129, 340139, 340160, 340535, 340598, 340971, 341039, 341457, 341636, 341737,
342011, 342353, 342374, 342683, 342807, 342963, 342996, 343009, 343158, 343201, 343291, 343598, 343655, 343847, 343968, 343969, 344073, 344196, 344259,
344338, 344399, 345019, 345040, 345062, 345114, 345390, 345781, 345835, 345897, 346026, 346161, 346178, 346250, 346562, 346678, 346712, 346950, 347167,
347407, 347511, 347655, 347908, 347972, 348203, 348277, 348382, 348580, 348639, 348794, 349007, 349038, 349144, 349170, 349212, 349386, 349408, 349663,
350280, 350302, 350500, 350765, 350819, 351134, 351328, 351397, 351566, 351686, 351719, 351759, 351807, 352312, 352357, 352389, 352651, 352814, 352821,
353146, 353200, 353284, 353607, 353893, 353999, 354391, 354445, 354525, 354631, 354690, 354716, 354738, 354791, 355119, 355159, 355223, 355440, 355571,
355593, 355621, 355697, 355717, 355779, 355863, 355922, 356374, 356535, 356569, 356665, 356702, 356916, 356922, 357010, 357272, 357289, 357340, 357508,
357790, 357877, 358029, 358033, 358134, 358223, 358239, 358253, 358289, 358405, 358543, 358706, 358714, 358770, 358788, 358789, 359308, 359323, 359357,
359865, 359868, 360017, 360555, 360570, 360585, 360719, 360759, 360811, 361197, 361685, 361939, 362157, 362247, 362657, 362699, 363190, 363252, 363331,
363363, 363593, 363602, 363671, 363719, 364455, 364467, 364468, 364719, 364913, 365015, 365138, 365314, 365427, 365464, 365659, 365729, 365739, 366009,
366071, 366148, 366313, 366430, 366480, 366795, 366956, 367164, 367357, 367549, 367630, 367716, 367792, 367934, 368363, 368589, 368637, 368833, 369016,
369509, 369735, 369801, 370124, 370152, 370400, 370505, 370524, 370537, 370728, 370790, 370831, 370986, 371029, 371361, 371392, 371486, 371786, 371824,
371871, 371923, 371955, 371960, 372003, 372112, 372156, 372247, 372292, 372309, 372319, 372352, 372558, 372669, 372748, 372871, 373393, 373444, 373645,
373653, 373727, 373730, 374340, 374391, 374818, 375245, 375294, 375331, 375380, 375820, 375996, 376090, 376241, 376258, 376454, 376573, 376750, 376802,
376819, 376838, 376848, 376941, 377007, 377513, 377518, 377570, 377609, 377926, 378586, 378775, 378791, 378916, 379034, 379093, 379136, 379315, 379349,
379434, 379820, 379853, 380122, 380395, 380429, 380440, 380885, 380889, 380949, 381128, 381961, 382005, 382069, 382341, 382469, 382472, 382514, 382620,
382784, 383576, 383605, 383660, 383807, 383917, 383929, 384691, 384745, 384836, 384917, 385107, 385337, 385401, 385704, 385882, 386154, 386211, 386401, 386784,
386934, 387105, 387124, 387202, 387256, 387293, 387338, 387365, 387717, 387849, 388403, 388421, 388469, 388894, 388961, 389145, 389154, 389157, 389292,
389498, 389705, 389743, 389772, 390310, 390365, 390474, 390565, 390567, 390663, 390969, 391063, 391175, 391229, 391332, 391439, 391488, 391733, 392167,
392180, 392362, 392657, 392684, 392869, 393095, 393325, 393608, 393924, 394151, 394495, 395013, 395211, 395221, 395259, 395271, 395425, 395432,
395791, 396014, 396193, 396380, 396536, 396784, 396825, 396933, 397212, 397217, 397390, 397423, 397569, 397760, 398036, 398172, 398305, 398712, 398729,
398872, 398901, 399208, 399276, 399354, 399408, 399432, 399442, 399835, 399922, 400124, 400534, 400740, 401001, 401269, 401455, 401962, 402020, 402041,
402042, 402264, 402298, 402575, 402632, 402806, 402946, 403133, 403197, 403535, 403705, 403841, 403888, 404183, 404205, 404270, 404473, 404475, 404852,
404899, 405013, 405136, 405520, 405582, 405604, 405663, 405709, 406121, 406187, 406230, 406295, 406328, 406988, 407038, 407173, 407246, 407460, 407688,
408163, 408664, 409166, 409454, 409488, 409678, 409754, 409824, 409825, 409918, 410024, 410107, 410165, 410707, 410708, 410916, 410963, 410969, 410992,
411104, 411191, 411238, 411289, 411446, 411778, 411803, 411862, 411877, 412002, 412194, 412691, 412756, 412910, 413088, 413615, 414002, 414850, 415235,
415529, 416076, 416117, 416286, 416355, 416651, 416907, 416948, 417070, 417141, 417220, 417365, 417832, 417844, 418056, 418065, 418500, 418717, 419001,
419019, 419026, 419028, 419110, 419324, 419396, 419599, 419664, 420363, 420366, 420823, 420864, 420892, 421059, 421298, 421596, 422029, 422064, 422255,
422367, 422583, 422782, 422969, 423250, 423341, 423711, 423806, 424049, 424161, 424193, 424278, 424376, 424408, 424485, 424694, 424844, 425052,
425325, 425628, 425721, 425825, 425945, 426383, 426510, 426551, 426705, 426728, 426829, 426838, 426849, 426877, 426979, 426988, 427169, 427238, 427301,
427362, 427435, 427555, 427628, 427654, 427756, 427779, 427805, 427852, 427853, 427920, 428093, 428576, 428787, 429059, 429437, 429536, 429594, 429959,
430563, 430731, 430889, 430925, 431112, 431178, 431211, 431376, 431704, 431328, 432598, 432615, 432683, 432754, 432897, 433240, 433296, 433398, 433662,
433921, 434201, 434951, 435029, 435272, 435453, 435471, 435869, 436025, 436168, 436620, 436797, 436941, 437080, 437224, 437547, 437632, 438045, 438071,
438099, 438292, 438331, 438429, 438462, 438478, 438663, 438795, 439060, 439273, 439303, 439374, 439509, 439692, 439765, 439889, 439906, 439991, 440002,
440313, 440389, 440511, 440614, 440820, 441205, 441599, 441640, 442062, 442298, 442461, 442680, 443136, 443410, 443455, 443505, 443527, 443562, 443725,
443944, 444033, 444036, 444285, 444344, 444346, 445127, 445276, 445323, 445392, 445405, 445462, 445540, 445689, 446303, 446383, 446539, 446670, 446677,
446726, 446864, 447179, 447424, 447457, 447574, 447681, 447934, 448115, 449158, 449414, 449469, 449780, 450162, 450551, 450707, 450878, 450914, 451119,
451283, 451336, 451337, 451482, 451800, 451818, 451842, 452014, 452229, 452524, 452565, 452619, 452837, 452873, 453002, 453137, 453311, 453553, 453620,
453704, 453906, 453930, 454144, 454174, 454246, 454258, 454406, 454570, 455313, 455406, 455424, 455543, 455649, 455667, 455677, 456176, 456554, 456658,
457085, 457190, 457225, 457286, 457555, 457660, 457976, 458057, 458124, 458172, 458633, 458751, 458762, 458827, 459346, 459465, 459747, 459951, 460139,
460228, 460370, 460442, 461099, 461908, 461996, 462067, 462383, 462445, 462523, 463224, 463338, 463417, 463474, 463505, 463953, 464166, 464174,
465101, 465200, 465457, 465829, 466024, 466223, 466242, 466523, 466885, 467774, 467905, 468117, 468219, 468401, 468465, 468518, 468760, 469293, 469427,
469545, 469559, 469567, 469658, 469825, 469832, 470004, 470072, 470085, 470174, 470393, 470501, 470893, 470976, 470977, 471096, 471136, 471315, 471332,
471665, 471698, 471962, 471966, 472506, 472654, 472749, 472954, 473003, 473072, 473348, 473352, 473373, 473500, 473879, 474123, 474342, 474424, 474461,
474699, 474725, 475007, 475037, 475129, 475236, 475533, 475651, 475754, 475988, 476060, 476347, 476934, 477005, 477040, 477266, 477392, 477580, 477590,
477797, 478105, 478148, 478164, 478712, 478833, 478892, 478899, 479168, 479172, 479396, 479707, 479886, 480014, 480196, 480240, 480576, 480741, 480779,
480843, 480908, 481165, 481218, 481292, 481355, 481428, 481530, 481736, 481804, 482093, 482215, 482326, 482330, 482454, 482472, 482706,
482731, 482775, 483015, 483078, 483261, 483363, 483611, 483766, 484108, 484206, 484208, 484307, 484369, 484385, 484620, 485016, 485306, 485364, 485367,
485602, 485632, 485705, 485757, 485800, 485954, 485984, 486300, 486606, 486713, 486936, 487009, 487264, 487284, 487464, 487502, 487510, 487602, 487806,
487992, 488073, 488139, 488404, 488641, 489107, 489145, 489695, 489971, 490056, 490097, 490087, 490180, 491204, 491249, 491273, 491477,
491666, 491707, 491739, 492040, 492138, 492155, 492219, 492268, 492293, 492302, 492408, 492638, 492894, 493504, 493626, 493936, 494032, 494128, 494174,
494257, 494534, 494706, 495233, 495528, 495776, 496053, 496261, 496374, 496457, 496732, 496752, 496839, 497296, 497311, 498297, 498508, 498639, 498669,
498679, 498706, 498730, 498854, 499122, 499141, 499155, 499538, 499682, 499798, 499966, 500057, 500214, 500440, 500561, 500594, 500603, 500686, 501269,
501710, 501842, 502015, 502134, 502148, 502300, 502407, 502470, 502553, 502679, 502726, 503022, 503478, 503497, 503500, 503541, 503777, 504211, 504259,
504554, 504616, 504744, 504748, 504769, 505288, 505884, 505885, 505895, 505898, 505924, 506030, 506056, 506162, 506226, 506231, 506592, 506740, 506837,
507073, 507215, 507266, 507342, 507642, 507761, 507765, 507776, 507815, 507952, 508140, 508200, 508311, 508429, 508456, 508467, 508504, 509039, 509269,
509555, 509652, 509746, 510342, 510493, 510572, 510617, 510860, 510976, 510977, 511036, 511146, 511580, 511642, 511930, 511967, 512282, 512400, 512458,
512644, 512658, 512951, 513221, 513683, 513811, 514064, 514213, 514230, 514243, 514295, 514391, 514435, 514559, 514622, 515229, 515401, 515470, 515512,
515590, 515623, 515702, 515815, 516106, 516263, 516481, 516791, 516906, 516990, 517403, 517451, 517492, 517685, 517805, 517869, 517920, 517985, 518215,
518552, 518785, 518918, 518966, 519205, 519477, 519616, 519626, 520100, 520112, 520199, 520272, 520445, 520456, 520479, 520767, 520883, 521064, 521184,
521338, 521366, 521437, 521514, 521618, 521709, 522062, 522074, 522191, 522240, 522288, 522342, 522365, 522416, 522423, 522462, 522465, 522704, 522771,
522947, 523484, 523487, 523561, 523577, 523795, 523995, 524155, 524227, 524340, 524369, 524476, 524520, 524662, 524866, 524966, 524991, 525342, 525555,
526029, 526070, 526521, 526523, 526552, 526597, 526695, 526713, 526769, 526912, 527073, 527139, 527267, 527345, 527597, 527796, 527822, 527925, 528020,
528071, 528198, 528493, 528851, 528941, 528970, 528992, 529016, 529345, 529376, 529624, 530097, 530132, 530406, 530629, 530903, 531201, 531277, 531388,
531444, 531550, 531752, 531834, 532335, 532376, 532595, 532622, 532711, 533050, 533218, 533220, 533827, 533897, 534037, 534155, 534166, 534224, 534311,
534440, 534543, 534711, 535049, 535101, 535218, 535229, 535234, 535289, 535358, 535418, 535561, 535666, 535874, 536039, 536127, 536146, 536244, 536278,
536555, 536576, 536730, 536820, 536823, 536902, 536960, 537371, 537461, 537553, 537720, 537770, 537807, 537960, 538196, 538263, 538398, 538537, 538544,
538574, 538633, 538737, 538805, 539158, 539475, 539632, 539851, 539941, 540370, 540436, 541255, 541338, 541472, 541938, 541949, 542027, 542160, 542173,
542442, 542718, 542799, 542936, 542988, 543233, 543617, 543642, 543790, 543833, 543838, 543882, 543947, 544109, 544169, 544294, 544831, 544875, 545022,
545145, 545213, 545214, 545260, 545325, 545351, 545411, 545721, 545793, 545850, 546046, 546093, 546218, 546242, 546408, 547055, 547165, 547315, 547348,
547411, 547974, 548136, 548416, 548575, 548772, 549127, 549366, 549599, 549605, 550129, 550134, 550140, 550308, 550311, 550354, 550532, 550726, 550844,
550972, 551172, 551197, 551244, 551316, 551472, 551524, 551607, 551793, 551814, 551869, 551873, 552184, 552199, 552272, 552291, 552549, 553126, 553176,
553308, 553498, 553586, 554031, 554168, 554598, 554699, 554703, 554950, 555022, 555120, 555654, 555771, 555794, 556162, 556176, 556399, 556424, 556544,
556698, 556830, 556888, 557602, 557628, 557678, 557694, 557746, 558276, 558372, 558804, 558824, 558890, 559132, 559267, 559271, 559497, 559618, 559700,

559760, 559830, 560152, 560155, 560180, 560372, 560517, 560532, 560576, 560809, 560909, 561028, 561339, 561384, 561454, 561479, 561582, 561593, 561818, 562063, 562092, 562100, 562162, 562826, 563110, 563364, 563525, 563545, 563658, 564063, 564228, 564271, 564349, 564449, 564508, 564823, 565220, 565243, 565476, 565608, 565664, 565769, 565884, 566245, 566301, 566319, 566592, 566612, 566968, 567189, 567396, 567566, 567937, 567964, 568272, 568654, 568725, 568788, 568840, 568851, 569214, 569234, 569255, 569261, 569286, 569769, 569795, 569851, 570178, 570211, 570285, 570581, 570656, 570878, 571563, 571654, 571658, 571661, 571671, 571694, 571702, 571719, 572179, 572307, 572353, 572529, 572602, 572689, 572732, 572786, 572801, 572949, 572998, 573476, 573632, 573704, 573825, 573875, 573961, 574251, 574299, 574368, 574420, 574443, 574563, 574760, 574870, 574957, 574961, 574983, 575049, 575417, 575461, 575519, 575980, 576153, 576157, 576286, 576543, 576581, 576829, 577126, 577140, 577197, 577399, 577405, 577416, 577558, 577583, 577725, 577850, 578002, 578294, 578369, 578519, 578523, 578567, 578702, 578805, 578808, 579057, 579255, 579299, 579440, 579667, 580008, 580238, 580374, 580668, 580695, 580785, 580905, 580957, 581282, 581518, 581563, 581766.

