# OpenReview forum: "LocCa: Visual Pretraining with Location-aware Captioners"
_NeurIPS.cc/2024/Conference — NeurIPS 2024 poster_

### Official Review · Reviewer_HN67 · 2024-07-09

**Soundness:** 4
**Presentation:** 3
**Contribution:** 3
**Rating:** 7
**Confidence:** 5

**Summary:**

This paper explored integrating region captions into caption-based language-image pertaining. Specifically, in addition to the autoregressive text modeling on global captions, the proposed LocCa also predicts constructed strings of "{bounding box}-{region caption}" and "{region caption}-{bounding box}". Although the method is conceptually simple, the author benchmarked LocCa in a variety of tasks and showed that it could achieve strong performance.

**Strengths:**

- The methodology is straightforward and elegant. The proposed LocCa unifies global and region-level caption generation and doesn't require additional complex architectural modifications.

- The benchmarking of downstream evaluation is quite comprehensive and the performance is strong.

**Weaknesses:**

1. **Objectives of LocCa**. Overall, the paper is well-written and easy to follow. However, the presentation on LocCa objectives in section 3.2 is a bit confusing starting from line 131. The "dual-faceted" loss is hard to interpret for first-time readers. If I understand correctly, it might mean that LocCa calculates loss on both box $b$ and region caption $c$. The author may consider rewriting this part to improve readability. Also, citing references regarding "contrasting with traditional approaches" could help readers understand the novelty. Additionally, how to construct a batch with three tasks should also be mentioned somewhere in section 3.

1. **Box predictions in GCap**. In the grounded captioning (GCap) task, conditioned on the image, the model needs to first predict a bounding box, and then predict the caption. The task of predicting a box from the image looks like an object proposal generation task, but the supervision here is only one box per sample. However, it's not clear whether introducing this supervision of object proposal generation is desired by the authors since there are no relevant discussions.

1. **Ablation of pretraining tasks**. Table 6 is very important for this paper. The authors ablated AREF and GCAP tasks respectively and showed that it leads to performance drops. To make the paper more complete, the authors may consider adding the ablation of only removing the box part and keeping the prediction to the region captions. This will help demonstrate the value of location awareness. Also, ablating the global caption and keeping AREF and/or GCAP may also be interesting.

1. **String-based box representation**. Compared to special tokenizations for box coordinates, whether the string-based method require more tokens to represent one bounding box? Whether it lead to less efficient decoding? Also, the hyperparameter of box coordinates resolution also plays an important role here and needs to be discussed and studied with ablations. The authors may see the discussion and explorations in the following papers for reference on bounding box representations:
    - [1] Shikra: Unleashing Multimodal LLM’s Referential Dialogue Magic
    - [2] Kosmos-2: Grounding Multimodal Large Language Models to the World
    - [3] RemoteCLIP: A Vision Language Foundation Model for Remote Sensing

**Questions:**

N/A

---

> ### Author Rebuttal · Authors · 2024-08-06
>
> 1. **Weaknesses 1: Objectives of LocCa**
>
>     Thanks for pointing out the "dual-faceted" loss. The understanding is correct, and we will revise this section for clarity. By "traditional approaches," we are referring to methods like OFA [1] and Florence-2 [2], which we will appropriately cite in the final manuscript. Additionally, we have explained how to construct a batch with three tasks in Lines 163-166, and we will include more details in Section 3 as suggested.
>
> 2. **Weaknesses 2: Box predictions in GCap**
>
>     The introduction of object proposal generation supervision in the GCap task is indeed an intentional design choice. This approach is conceptually similar to the first RoI prediction in the Pix2Seq model [3]. Our hypothesis is that providing partial location information (such as the coordinates of the upper-left corner of an object) enables the model to predict the complete bounding box coordinates. This assumption justifies the introduction of additional supervision as a reasonable enhancement to the model’s learning process.
>
>     As detailed in Section 4.3 and Appendix C, one of the additional benefits of this design is the flexibility it offers during the prediction phase. For instance, we can input a specific coordinate to prompt the model to generate a corresponding regional caption. Alternatively, we can simply input the task prefix ('GCAP') and use beam search to allow the model to output various detection results. This flexibility enriches the model’s utility by accommodating different levels of input specificity, thereby enhancing its practical applicability in diverse scenarios.
>
> 3. **Weaknesses 3: Ablation of pretraining tasks**
>
>     - We understand your intention and appreciate the suggestion regarding the ablation of the bounding box component. However, the integration of spatial and textual information is central to the LocCa model's design. Removing the box component would impede the model’s ability to learn about spatial coordinates, which are crucial for its zero-shot prediction capabilities and understanding the spatial context of objects. The LocCa framework is specifically developed to link region box locations with captions to enhance cross-modal associations. Thus, omitting the bounding box data would significantly diminish the model’s ability to comprehensively understand visual scenes. Therefore, we have focused our evaluations on scenarios that maintain the full scope of the model’s cross-modal learning functionality.
>     - We totally agree it would be interesting to see the impact of the global caption (Cap) task. The experimental setup is similar to that in Table 6 in the paper, but we used a larger decoder with 12 layers. Comparing Exp1 and Exp2 in **Table 3 in the rebuttal PDF file**, we observed a consistent performance drop without the Cap task on both RefCOCOs and holistic tasks. The drop was more pronounced on holistic tasks like COCO captioning and image classification, suggesting that the Cap task is beneficial for holistic image representation learning.
>
> 4. **Weaknesses 4: String-based box representation**
>
>    - String-based tokenization requires more tokens than special tokenization, making it less efficient. For instance, using the c4 tokenizer [4], '498' requires two tokens for string representation but only one token with special tokenization.
>
>     - As shown in the **Table 3 in the rebuttal PDF file**, by comparing Exp 3,1,4, we did not observe a noticeable difference in performance across various box coordinate resolutions (224, 500, 1000) with image resolution of 224, indicating that LocCa remains robust despite changes in coordinate resolution. We set the coordinate resolution to 500 to adapt to different image resolutions (224 -> 384 -> 640), as shown in Figure 3. We believe that further lowering the coordinate resolution (such as 32 that much less than 224) could degrade the model’s accuracy, as each coordinate might then correspond to multiple pixels, leading to potential mapping errors.
>
>     - Thanks for providing Shikra, Kosmos-2, and RemoteCLIP as references for bounding box representations. We compared these with LocCa and noted distinct application scenarios. Shikra explored both string and special tokenization methods, ultimately opting for string tokenization, which we also employed. Shikra uses a LLM and a frozen visual encoder,  making it difficult to train a new coordinate vocabulary that aligns with the pretrained VL representation. In contrast, LocCa trains the entire model from scratch, allowing for easier adaptation. This explains the significant performance gain of around 2% on the RefCOCO+/g with string tokens in Shikra, but the gap in LocCa is much smaller (<0.3%). Additionally, Kosmos-2 introduces location tokens for each image grid rather than the coordinates. This allows a bounding box to be represented by just two tokens, enhancing efficiency. However, we believe these tokens are too coarse, potentially impairing the model’s ability to accurately learn and represent small objects. RemoteCLIP suggests converting the box coordinates into text strings and concatenating them with other textual data before encoding them with the text encoder. It avoids using complex region-text alignment (like RegionCLIP). However, without explicitly constructing the correspondence between regional visual features and box coordinates, achieving fine-grained region-wise alignment may be challenging.
>
> Reference:
>
> [1] Wang et. al. OFA: Unifying Architectures, Tasks, and Modalities Through a Simple Sequence-to-Sequence Learning Framework.
>
> [2] Xiao et. al. Florence-2: Advancing a Unified Representation for a Variety of Vision Tasks.
>
> [3] Chen et al. Pix2seq: A Language Modeling Framework for Object Detection.
>
> [4] https://github.com/mosaicml/streaming/

---

> ### Comment · Reviewer_HN67 · 2024-08-08
>
> Thanks for the response and updated experiments. They are all sound to me. Please make sure the mentioned revisions are incorporated into the final version.
>
> I have updated my rating from weak accept to accept.

---

> > ### Author Response · Authors · 2024-08-12
> >
> > Thanks for your recognition of our work and for raising the score! We will add the mentioned revisions in the final manuscript.

---

### Official Review · Reviewer_ris8 · 2024-07-12

**Soundness:** 3
**Presentation:** 3
**Contribution:** 3
**Rating:** 7
**Confidence:** 4

**Summary:**

The paper introduces LocCa, a new visual pretraining approach that incorporates location-specific tasks into image captioning-based vision language models, improving their ability to extract detailed information from images. The authors propose two location-aware tasks, automatic referring expressions (AREF), which predicts bounding box coordinates from captions, and grounded captioning (GCAP), to jointly predict the box coordinates and captions from image, utilizing a multitask encoder-decoder architecture. As a result, LocCa significantly outperforms standard captioning models on localization challenges, achieving top results on datasets such as RefCOCO/+/g, while maintaining similar performance on broader tasks.

**Strengths:**

- The paper is clear and well-written.
- The proposed approach demonstrates good performance on a broad number of downstream tasks.
- The paper provides a wide range of analyses.

**Weaknesses:**

- The model's training process relies on regional captions and bounding boxes, necessitating the use of pre-existing tools like OWL-ViT to produce detailed object locations for its training data. This approach creates a significant dependency on these off-the-shelf models, potentially limiting the quality of the training data. Assembling a large-scale dataset becomes challenging under these constraints. While the authors have managed to curate an impressive billion-scale dataset, this quantity doesn't necessarily ensure data quality. The reliance on external models for data generation raises questions about the overall integrity and reliability of the training information.

- Typo
    - Line 134: AREFtask → AREF task
    - Line 240: dentified → identified

**Questions:**

- I have two questions about the proposed training tasks.
    - During the pretraining process or after training, is there any trend related to the size of the object in the image? Such analysis can be done among downstream tasks, such as object detection or RefCOCO series datasets.
    - Can you provide more details on the detection visualization analysis presented in Appendix C? Specifically, what is the connection between observing only one object per example and the substantial bounding box overlap observed without Non-Maximum Suppression (NMS)? Additionally, doesn't the LocCa pretraining process involve comprehension of multiple objects through its three tasks: captioning, AREF, and GCAP? How does this align with the single-object observation?

**Limitations:**

The paper addresses the limitations.

---

> ### Author Rebuttal · Authors · 2024-08-06
>
> 1. **Weaknesses: Data quality and dependency on external models & Typos**
>
>     - To address the concerns about data quality and dependency on external models like OWL-ViT, we would like to emphasize our strategic use of simple filtering techniques to enhance data quality. Specifically, we employ a confidence score threshold of greater than 0.3 for selecting bounding boxes. This threshold is chosen based on its proven effectiveness in balancing accuracy and inclusivity, thus minimizing the inclusion of erroneous or irrelevant data. This filtering approach is similar to the use of the CLIP score in creating high-quality image-text datasets like LAION-400M [1] and YFCC15M [2], and its effectiveness has also been demonstrated in OWL-ViT-2 [3].
>
>     - Thanks for pointing out the typos; we will correct them in the final manuscript.
>
> 2. **Q1: Analysis on object size**
>
>     We have analyzed the prediction results related to box size in the MSCOCO-DET dataset. The table below presents the mean Average Precision (mAP) for different box sizes, using the widely accepted classifications: large (>96$^2$ pixels), medium (32$^2$ - 96$^2$ pixels), and small (<32$^2$ pixels). We found that larger boxes consistently achieve higher mAP, a trend echoed in other object detection studies such as Pix2Seq [4] and SSD [5]. This pattern holds across various visual backbones, including CLIP, Cap, CapPa, and LocCa. The higher mAP for larger boxes can be attributed to the more detailed visual features they provide, making them easier for models to detect.
>
>
>     | Model        | mAP   | mAP-small | mAP-medium | mAP-large |
>     |-----------------|:------------:|:-----------:|:------------:|:-----------:|
>     | CLIP  | 40.46 | 19.11     | 44.46      | 62.79     |
>     | Cap          | 39.00 | 18.31     | 42.56      | 60.55     |
>     | CapPa        | 39.48 | 18.32     | 43.21      | 59.81     |
>     | LocCa        | 47.73 | 26.88     | 52.80      | 67.64     |
>
>
> 3. **Q2: Visualization details**
>
>     During the LocCa pretraining process, each example specifically sampled one box-text pair for the AREF and GCAP tasks, individually (we also explored using multiple box-text pairs for each task during pre-training, it did not enhance performance and instead increased computational costs). We hypothesize that through multiple epochs of training, LocCa can observe all candidate box-text pairs in the training set.
>
>     When deploying the pretrained model for zero-shot predictions with the GCAP task, it executes multiple decodings for a single input image using beam search, as described in Section 4.3. By setting the beam search to a beam size of 1 (single beam) and a low temperature, the model consistently selects the most confident bounding box regions. While this sampling strategy yields high-confidence predictions, it also results in substantial bounding box overlaps without the application of Non-Maximum Suppression (NMS).
>
> Reference:
>
> [1] Schuhmann et. al. LAION-400M: Open Dataset of CLIP-Filtered 400 Million Image-Text Pairs.
>
> [2] https://huggingface.co/datasets/mehdidc/yfcc15m
>
> [3] Minderer et.al. Scaling Open-Vocabulary Object Detection.
>
> [4] Chen et al. Pix2seq: A Language Modeling Framework for Object Detection.
>
> [5] Liu et al. SSD: Single Shot MultiBox Detector.

---

> > ### Author Response · Authors · 2024-08-12
> >
> > Dear reviewer,
> >
> > Thank you again for the detailed and constructive comments.
> >
> > As the discussion period is about to end, we would like to ensure we've addressed all of your questions and concerns. If you feel we have satisfactorily responded, please let us know. Otherwise, please let us know your remaining concerns so we can address them before the discussion period closes.
> >
> > Sincerely

---

> ### Comment · Reviewer_ris8 · 2024-08-13
> **Final rating**
>
> Thank you for the rebuttal. As most of my concerns and questions are addressed, I will maintain my initial rating.

---

### Official Review · Reviewer_gCZj · 2024-07-16

**Soundness:** 3
**Presentation:** 3
**Contribution:** 3
**Rating:** 5
**Confidence:** 5

**Summary:**

This work proposed a location-aware pre-training for vision-language learning. The pre-training contains two tasks: one has location input and output caption/text; the other one has text input and output location of the corresponding object. The model is trained from scratch in ~1B large-scale image-text pairs with object location extracted by an off-the-shelf detection model. The experiments demonstrate the strong performance of the proposed model and the generalizability of the pre-trained vision encoder.

**Strengths:**

1. The experiment section covers a wide range of tasks including location-aware tasks and holistic vision-language tasks. The ablation is also comprehensive. It's also impressive that the pre-trained vision backbone can be used as a general visual encoder to replace other CLIP-like models.

2. The model is lightweight. In the era of scaling up models, the proposed model can achieve great performance with just 600+M parameters.

**Weaknesses:**

1. The key novelty is claimed to be pre-training a vision-language model via two location-aware tasks (Line139): `automatic referring expression` and `grounded captioning`. However, similar location-input-text-output and text-input-location-output tasks have been widely used in the training (mostly fine-tuning though) of location-aware MLLMs recently, eg, Shikra[1], Ferret[2], GLaMM[3]. Moreover, Ferret-v2 [4] also proposed dense referring and dense grounding as a pre-training stage. Those two tasks are quite similar to what this work proposes, and they even involve multiple regions in one round.

2. Missing comparison with many location-aware Multimodal LLM methods.  Location-aware MLLMs load pre-trained LLMs and train on a moderate amount of data (100k-1M) in a few steps (mostly within 3 epochs). They can already show great performance in basic tasks and reasoning tasks. This work instead trains the model from scratch with large-scale data (1B) with longer training time. The authors should analyze the benefits and drawbacks of each line of work.

3. Missing Evaluations on: (1). Location-aware Reasoning tasks, such as Visual-7W[5], LookTwice-QA[6] used in Shikra[1] and Ferret-Bench used in Ferret[2]. (2). Grounded captioning capability is not evaluated, for example, Flickr30k used in GLaMM and Ferret. (3). Comparison with Multimodal LLMs in RefCOCOs and above-mentioned tasks.

4. Training data, WebLI dataset, is not available to the public. Considering the main novelty, location-aware pre-training, largely depends on the scale and quality of pre-training dataset, the work is hard to be reproduced.


Refs: \
[1] Chen, Keqin, et al. "Shikra: Unleashing multimodal llm's referential dialogue magic." arXiv preprint arXiv:2306.15195 (2023). \
[2] You, Haoxuan, et al. "Ferret: Refer and ground anything anywhere at any granularity." ICLR 2024 \
[3] Rasheed, Hanoona, et al. "Glamm: Pixel grounding large multimodal model." CVPR. 2024. \
[4] Zhang, Haotian, et al. "Ferret-v2: An Improved Baseline for Referring and Grounding with Large Language Models." arXiv preprint arXiv:2404.07973 (2024). \
[5] Zhu, Yuke, et al. "Visual7w: Grounded question answering in images." Proceedings of the IEEE conference on computer vision and pattern recognition. 2016.
[6] Mani, Arjun, et al. "Point and ask: Incorporating pointing into visual question answering." arXiv preprint arXiv:2011.13681 (2020).

**Questions:**

Please see the weaknesses.

=====================

Thank the author for the answers. Please make sure to add the experiments of Visual7W and LookTwice-QA as promised. I'd like to raise my rating.

**Limitations:**

Yes, It is discussed in the appendix.

---

> ### Author Rebuttal · Authors · 2024-08-06
>
> 1. **Weaknesses 1&2: Comparison with location-aware MLLMs**
>
>     Thanks for highlighting the connections with existing location-aware MLLMs like Shikra, Ferret, and GLaMM which we will cite in our final manuscript. We discuss the differences between LocCa and the referenced work as follows:
>
>     - Firstly, we appreciate the opportunity to clarify the fundamental objectives and innovations of our work with LocCa, as discussed in Sec. 3.3 of our manuscript. Our primary goal with LocCa is not to enhance MLLMs with location awareness as described in the referenced works. These works typically utilize a pretrained CLIP image encoder and incorporate location-aware tasks during either the fine-tuning stages or as secondary enhancements to already trained models. In contrast, LocCa is a new approach to visual pretraining that inherently integrates location-aware tasks from the very beginning.
>     - Besides, LocCa modifies the standard REF and GCAP tasks to better accommodate this integration, as detailed in Lines 130-137 in the paper. As a complementary, the visual encoder developed through LocCa can be easily incorporated into any MLLM, as evidenced by its application in the PaLI-3 experiments shown in Table 4 in the paper.
>    - Ferret-v2 is a concurrent paper. We also investigated dense referring and dense grounding as proxy tasks during LocCa pretraining. The results were on par with AREF/GCAP, but required a significantly longer decoding length for the decoder, making the process more computationally intensive while providing little or no benefits.
>
>
> 2. **Weaknesses 3: Missing Evaluations**
>
>    - (1) Thanks for highlighting the missing tasks on location-aware reasoning. Both Visual7W and LookTwice-QA are relevant. In Visual7W, six types of questions (what, where, when, who, why, and how) assess a model’s visual understanding capabilities, akin to the VQA/GQA tasks we selected in Table 3 in the paper. The seventh question category (which) is similar to the task of REC. Therefore, we actually almost covered all the task types presented in Visual7W. Furthermore, Visual7W is derived from MS-COCO, which also underlies VQAv2 and RefCOCOs for which we present results in the paper.  LookTwice-QA requires situating a local region in the broader context of the image, which is meaningful while not widely used in recent research. **We plan to include both tasks in our future evaluations**.
>
>         Ferret-Bench typically necessitates a LLM for knowledge-based reasoning, which is not the main focus of LocCa. While integrating the pretrained LocCa visual encoder with a LLM (e.g., Vicuna used in Ferret) for complex tasks is feasible, it falls outside the scope of this work.
>
>    - (2) For grounded captioning, we clarify that GRIT/GPT4RoI/GLaMM evaluate on RefCOCO and Visual Genome, whereas Ferret evaluates on Flickr30k. The evaluation tasks differ between these two studies: the former involves generating regional captions based on a given bounding box location, while Ferret requires the model to generate a caption and provide bounding boxes for all noun phrases within the caption. Our setup aligns more closely with GRIT/GPT4RoI/GLaMM, so we report our grounded captioning results on Visual Genome and make comparisons with these methods.
>
>         As shown in the **Table 1 in the rebuttal PDF file**, the LocCa$_L$ model with only 0.6B parameters (without LLM), outperforms GPT4RoI-13B (with LLM) on both METEOR and CIDEr scores for the Visual Genome grounded captioning task. When compared with GLaMM, it performs better on the METEOR metric but lags on CIDEr. METEOR is an evaluation metric that focuses on the precision, recall, and alignment of words between the generated caption and the reference caption, while CIDEr evaluates the similarity of n-grams between the generated captions and the reference captions. Visual Genome has relatively simple captions, typically consisting of a few words. LocCa adopts a simpler decoder (0.3B params only) that closely matches the reference captions in terms of word choice and order. The simplicity of the decoder might limit its ability to produce complex sentences but enables it to generate concise captions that align well with VG's simple nature. GLaMM, on the other hand, uses a more complex LLM as its decoder (Vicuna-7B, 20x params compared to LocCa's decoder), which likely generates more diverse captions that include n-grams that match the reference captions more effectively. This could explain why LocCa has a higher METEOR score but a lower CIDEr score compared to GLaMM. However, it is important to note that the **pretrained LocCa encoder is complementary to MLLMs (i.e. as a better alternative option to the CLIP encoder)**, and we expect further performance gains on downstream tasks when combining both.
>
>    - (3) As shown in the **Table 2 in the rebuttal PDF file**, LocCa$_G$ achieves higher performance than MLLM (e.g. Shikra, Ferret) on RefCOCOs even with a small decoder rather than a LLM (1.3B vs 13B).
>
> 3. **Weaknesses 4: Reproduction**
>
>    - First, in our paper we only adopted a publicly available OWL-ViT CLIP L/14 model [7] to generate detection pseudo annotations. We believe a similar dataset (e.g. based on publicly accessible data) could be reproduced with the technical details provided and the public OWL-ViT models.
>
>    - In the meantime, as in other areas of large-scale science, we believe that publishing as many details as possible on the findings of state-of-the-art systems is beneficial to the community. We will also try our best to provide detailed descriptions on construction of the dataset, and model training methods.
>
>    - Finally, in our ongoing work, we are actively exploring training our models using only publicly accessible data, so that we can make a fast and accessible model. We are making our best effort to release these artifacts.
>
> Refs:
>
> [7] Minderer et. al. Simple Open-Vocabulary Object Detection with Vision Transformers.

---

> > ### Author Response · Authors · 2024-08-12
> >
> > Dear reviewer,
> >
> > Thank you again for the detailed and constructive comments.
> >
> > As the discussion period is about to end, we would like to ensure we've addressed all of your questions and concerns. If you feel we have satisfactorily responded, please let us know. Otherwise, please let us know your remaining concerns so we can address them before the discussion period closes.
> >
> > Sincerely

---

### Official Review · Reviewer_88rJ · 2024-07-16

**Soundness:** 3
**Presentation:** 3
**Contribution:** 2
**Rating:** 5
**Confidence:** 3

**Summary:**

This paper presents LocCa, a novel visual pretraining paradigm that incorporates location-aware tasks into captioners. Specifically, LocCa employs two tasks, bounding box prediction and location-dependent captioning, conditioned on the image pixel input. This multi-task training helps LocCa significantly outperforms standard captioners on downstream localization tasks while maintains comparable performance on holistic tasks.

**Strengths:**

1. The paper is well written. The core contributions are clearly presented.

2. The proposed method achieves state-of-the-art performance.

**Weaknesses:**

Several works also investigate he matching of image regions with corresponding text during pretraining. The authors claim that compared to these methods, the proposed LocCa can git rid of complex model architectures and become more computationally efficient. To demonstrate this, more experimental results should be provided to validate this claim, for example, FLOPs, trainable paramters or training time.

**Questions:**

Several works also investigate he matching of image regions with corresponding text during pretraining. The authors claim that compared to these methods, the proposed LocCa can git rid of complex model architectures and become more computationally efficient. To demonstrate this, more experimental results should be provided to validate this claim, for example, FLOPs, trainable paramters or training time.

**Limitations:**

Several works also investigate he matching of image regions with corresponding text during pretraining. The authors claim that compared to these methods, the proposed LocCa can git rid of complex model architectures and become more computationally efficient. To demonstrate this, more experimental results should be provided to validate this claim, for example, FLOPs, trainable paramters or training time.

---

> ### Author Rebuttal · Authors · 2024-08-06
>
> Previous work, such as RegionCLIP, has also attempted to align image regions with corresponding textual descriptions using contrastive pre-training schemes. However, these approaches tend to be resource-intensive and less efficient:
>
> - In terms of trainable parameters, RegionCLIP [1] uses a pre-trained CLIP from OpenAI [2] as a teacher model to train a student visual encoder of equivalent size. Besides, it incorporates an ROI-Align module for extracting regional features, which increases the complexity. Conversely, LocCa aligns with the Cap model in simplicity, having a similar number of parameters, i.e., about half the parameters of the RegionCLIP model, taking the teacher model into account.
>
> - In terms of training time, RegionCLIP requires extracting 100 RoIs per image and conducting dual-level contrastive learning—both at the image and region levels. This significantly increases the computational load per example by a factor of 100x|C| (where |C| is the average number of concepts per caption) during the contrastive learning stage. According to [3], with a pre-trained CLIP, training RegionCLIP (with RN50 encoder) takes 6 days using 32 V100 GPUs on 57.6M examples, which is 80k V100-hours or approximately 20k TPUv4-hours per billion examples [4][5]. Conversely, LocCa (B/16 encoder) reuses encoded visual features across 3 tasks, leading to an overall training time per billion examples that is only 1.3 times longer (611 vs 454 TPUv4-hours per billion examples) than that of Cap, which is much faster than RegionCLIP.
>
> | Model      | Params | TPUv4-hrs. |
> |------------|:--------:| :----------: |
> | B/16 Cap     | 192 M  | 454     |
> | B/16 CLIP*  | 197 M  | 444     |
> | B/16 LocCa | 192 M  | 611      |
> | R50 RegionCLIP | -  | 20000  |
>
>
> Reference:
>
> [1] Zhong et. al. RegionCLIP: Region-based Language-Image Pretraining.
>
> [2] https://github.com/openai/CLIP
>
> [3] https://github.com/microsoft/RegionCLIP/issues/50
>
> [4] https://lambdalabs.com/blog/nvidia-a100-vs-v100-benchmarks
>
> [5] https://cloud.google.com/blog/products/ai-machine-learning/google-wins-mlperf-benchmarks-with-tpu-v4

---

> > ### Author Response · Authors · 2024-08-12
> >
> > Dear reviewer,
> >
> > Thank you again for the detailed and constructive comments.
> >
> > As the discussion period is about to end, we would like to ensure we've addressed all of your questions and concerns. If you feel we have satisfactorily responded, please let us know. Otherwise, please let us know your remaining concerns so we can address them before the discussion period closes.
> >
> > Sincerely

---

### Author Rebuttal · Authors · 2024-08-07

We appreciate the valuable advice and generally positive feedback from all reviewers. Specifically, Reviewers 88rJ, ris8, and HN67 find our paper clear and well-written. Reviewer HN67 considers our methodology straightforward and elegant. Reviewers gCZj, ris8, and HN67 acknowledge that our experiment encompasses a wide range of downstream tasks. All reviewers agree that our performance is strong with a lightweight model.

We addressed the reviewers' concerns in a separate section. To provide detailed experimental results, we have included a PDF file containing three tables. Please refer to this PDF file, where the relevant information is indicated as **Table x in the rebuttal PDF file**.

---

### Decision · Program_Chairs · 2024-09-25

**Decision:**

Accept (poster)

**Comment:**

This paper presents a novel pretraining framework for vision foundation models. To enhance  location awareness of a model, the proposed method incorporates two location-related tasks, bounding box prediction and location-dependent captioning, in addition to the standard holistic captioning.
The reviewers appreciated the strong performance of the method on downstream tasks, extensive experiments, and good writing and presentation of the paper. Meanwhile, they also raised some concerns. Some major ones include; comparison and positioning of the proposed method with respect to other location-aware MLLMs, lack of evaluation on some downstream tasks, insufficient ablation studies, lack of evidence that the prosed method is more efficient than previous location-based pretraining  methods, etc.
During the discussion phase, the authors addressed the above concerns carefully with many new experimental results as evidence. The reviewers have been generally satisfied with the authors rebuttal, and two of them decided to upgrade the scores, resulting in unanimous recommendation for acceptance of the paper. The AC agrees that the reviewers’ concerns have been properly addressed and cleared, and concludes that the paper is above the bar for acceptance. The AC does encourage the authors to take in the new results, analysis and discussion in the camera ready version as promised.